# Gaussian Scenes: Pose-Free Sparse-View Reconstruction using Depth-Enhanced Diffusion Priors

**Soumava Paul, Prakhar Kaushik, Alan Yuille**
**CCVL, Johns Hopkins University**
`{spaul27, pkaushi1, ayuille1}@jhu.edu`

**Reviewed on OpenReview:** [https://openreview.net/forum?id=yp1CYo6R0r](https://openreview.net/forum?id=yp1CYo6R0r)

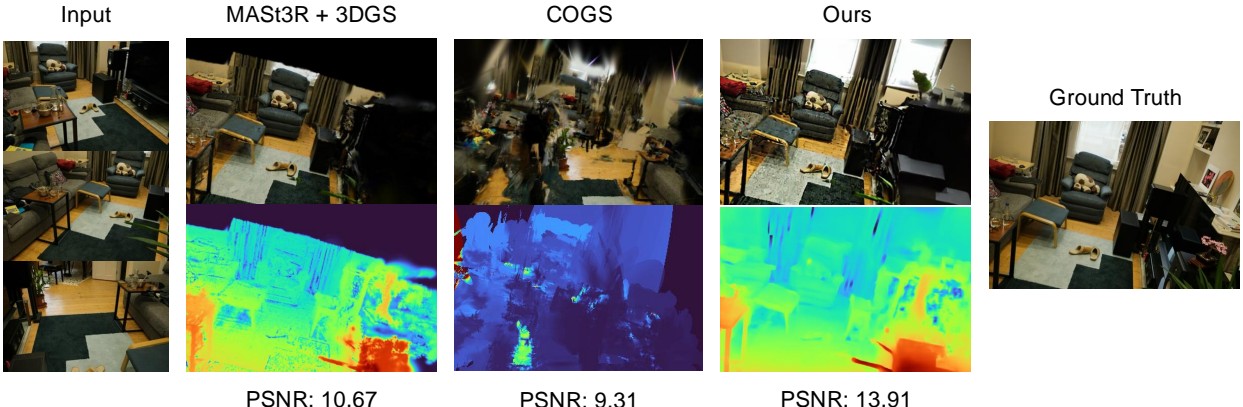

Figure 1: Given sparse *pose-free* images as input, *GScenes* reconstructs a 3D scene in 5 minutes by iteratively fusing novel view renders and depth maps with an underlying 3D Gaussian representation. Typical pose-free baselines built with geometric priors struggle with reconstructing 360°scenes from sparse inputs due to the absence of generative priors. *GScenes* comprises a latent diffusion model capable of inpainting missing details and removing Gaussian artifacts in novel view renders, thereby enabling generation of full 360°scenes.

## Abstract

In this work, we introduce a generative approach for *pose-free* (without camera parameters) reconstruction of 360°scenes from a sparse set of 2D images. Pose-free scene reconstruction from incomplete, pose-free observations is usually regularized with depth estimation or 3D foundational priors. While recent advances have enabled sparse-view reconstruction of large complex scenes (with high degree of foreground and background detail) with known camera poses using view-conditioned generative priors, these methods cannot be directly adapted for the pose-free setting when ground-truth poses are not available during evaluation. To address this, we propose an image-to-image generative model designed to inpaint missing details and remove artifacts in novel view renders and depth maps of a 3D scene. We introduce context and geometry conditioning using Feature-wise Linear Modulation (FiLM) modulation layers as a lightweight alternative to cross-attention and also propose a novel confidence measure for 3D Gaussian splat representations to allow for better detection of these artifacts. By progressively integrating these novel views in a Gaussian-SLAM-inspired process, we achieve a multi-view-consistent 3D representation. Evaluations on the MipNeRF360 and DL3DV-10K benchmark dataset demonstrate that our method surpasses existing pose-free techniques and performs competitively with state-of-the-art *posed* (precomputed camera parameters are

given) reconstruction methods in complex 360°scenes. Our project page[1] provides additional results, videos, and code.

# 1 Introduction

Reconstructing high-quality 3D scenes from sparse images remains a fundamental challenge in computer vision. While recent methods employ various priors to stabilize NeRFs (Mildenhall et al., 2020) or Gaussian splats (3DGS) (Kerbl et al., 2023) in under-constrained scenarios, they typically require accurate camera parameters derived from dense observations—a restrictive assumption for real-world applications. Pose estimation from sparse views is inherently challenging; both traditional Structure from Motion and recent foundational models (Wang et al., 2024; Leroy et al., 2024) struggle with insufficient matching features. Current pose-free 3DGS approaches integrate monocular depth (Ranftl et al., 2020), semantic segmentation (Kirillov et al., 2023), or 3D priors (Wang et al., 2024), but fail on complex 360°scenes with sparse coverage, highlighting the need for additional generative regularization.

Although most previous sparse view 3D scene reconstruction works are *posed* and require ground-truth camera poses (DSNeRF (Deng et al., 2022), DNGaussian (Li et al., 2024), SparseGS (Xiong et al., 2023), SparseN-eRF (Wang et al., 2023a), DietNeRF (Jain et al., 2021), FreeNeRF (Yang et al., 2023), RegNeRF (Niemeyer et al., 2022), DiffusioNeRF (Wynn and Turmukhambetov, 2023)), recent works like COGS (Jiang et al., 2024) and InstantSplat (Fan et al., 2024) have found initial success in the *pose-free* sparse view reconstruction problem. Although they have been successful in reducing 3D reconstruction-related artifacts such as blur, floaters, color, and streak-like artifacts, they lack generative capabilities for a complete 360°reconstruction, which requires robust priors from powerful generative models (Saharia et al., 2022; Rombach et al., 2022). In addition, most of these methods lack mechanisms to robustly identify and locate these artifacts in the reconstructed scenes. Recent methods like ZeroNVS (Sargent et al., 2024), Reconfusion (Wu et al., 2024), and CAT3D (Gao* et al., 2024) incorporate 3D view conditioning for realistic extrapolation but depend on accurate poses and cannot be trivially extended to the pose-free problem. Gaussian Object (Yang et al., 2024), iFusion (Wu et al., 2023), and UpFusion (Nagoor Kani et al., 2024) do provide pose-free generative solutions but remain limited to only object reconstruction rather than scene reconstruction. Other generative solutions like ZeroNVS, ReconFusion, and CAT3D use stronger priors over regularization-based techniques like FreeNeRF, RegNeRF, and DietNeRF but rely on large-scale 3D datasets, video datasets, and compute resources not available to the average researcher. Furthermore, works like ReconFusion and CAT3D remain closed-source with no access to their models, data or other reproducibility data, which limits their usage for the community. We summarize such existing methods and their use cases and attributes in Fig 2.

To alleviate these challenges, we present *GScenes*, an efficient approach using 3D foundational and RGBD (RGB image and depth) generative priors for pose-free sparse-view reconstruction of complex 360°scenes. We first estimate a point cloud and approximate camera parameters using MASt3R (Leroy et al., 2024), then jointly optimize Gaussians and cameras with 3DGS. Novel views generated from an elliptical trajectory fitted to the training views contain artifacts that our diffusion prior refines to further optimize the scene. We condition a Stable Diffusion UNet (Rombach et al., 2022) on estimated cameras, context, 3DGS renders, depth maps, and a confidence map capturing artifacts and missing details. Our proposed confidence

| Method | Pose-free | Open-source | Generative Priors | Scene Reconstruction |
|---|---|---|---|---|
| FreeNeRF, DietNeRF, RegNeRF, DN-Gaussian, SparseGS, SparseNeRF | ✗ | ✓ | ✗ | ✓ |
| DiffusioNeRF, ZeroNVS | ✗ | ✓ | ✓ | ✓ |
| ReconFusion, CAT3D | ✗ | ✗ | ✓ | ✓ |
| Gaussian Object, iFusion, UpFusion | ✓ | ✓ | ✓ | ✗ |
| InstantSplat, COGS | ✓ | ✓ | ✗ | ✓ |
| *GScenes* (Ours) | ✓ | ✓ | ✓ | ✓ |

Figure 2: **Comparison of sparse-view reconstruction methods.** Methods are grouped based on their requirement for accurate camera poses, open-source availability, need for generative priors, and applicability to large-scale scene reconstruction.

---

measure is designed by combining per-pixel light transmittance with Gaussian density, which provides a reliable conditioning signal to the Stable Diffusion UNet on the presence of empty regions and Gaussian artifacts in novel view renders and depth maps. For training our diffusion prior, we create our own dataset of 171, 461 samples using scenes from open-source multiview datasets. Each sample is a pair of clean RGBD images and images with empty regions and Gaussian artifacts, along with other conditioning signals (confidence map, clip features, etc). Augmented variants of both the Stable Diffusion VAE and UNet are finetuned using this synthetic dataset. Despite using weaker and cheaper generative priors than pose-dependent methods, we demonstrate competitive performance against closed source methods like ReconFusion (Wu et al., 2024) and CAT3D (Gao* et al., 2024) while outperforming other techniques without requiring million-scale multi-view or video data or extensive compute resources. Our contributions include:

- An image-to-image RGBD generative model for synthesizing plausible novel views from sparse pose-free images, using lightweight FiLM modulation (Perez et al., 2018) instead of cross-attention

- A confidence measure to detect artifacts in novel view renders, guiding our diffusion model toward effective novel view synthesis

- Integration of diffusion priors with MASt3R's geometry prior, enabling efficient scene reconstruction previously requiring 3D-aware video diffusion

- Superior performance compared to recent regularization and generative prior-based open-source methods for 3D scene reconstruction

- An open-source low-cost solution with lower data and compute requirements compared to state-of-the-art posed reconstruction methods.

See Appendix for further discussion and Related Works (Appendix B).

## 2 Method

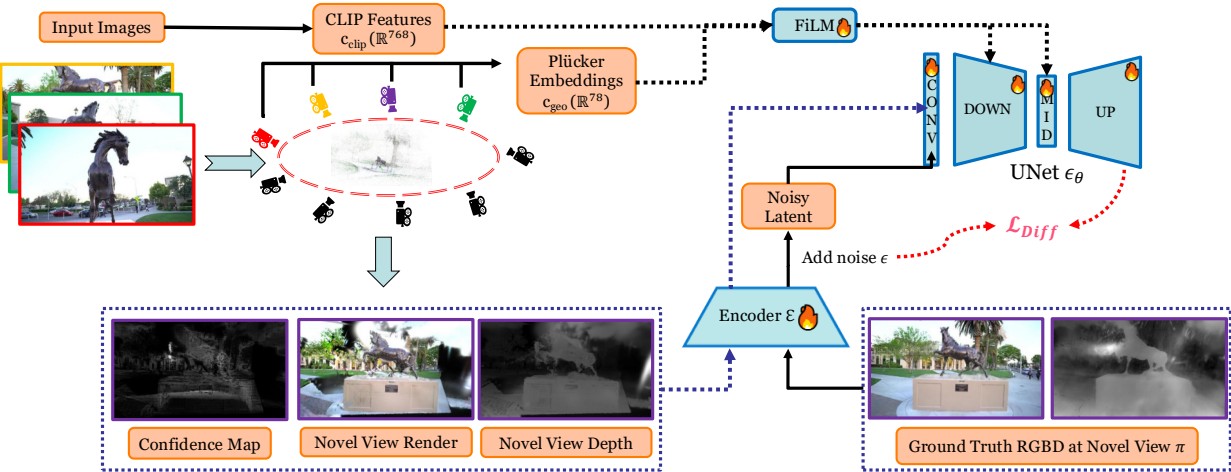

Figure 3: **Overview of *GScenes*.** We render 3D Gaussians fitted to our sparse set of $M$ views from a novel viewpoint. The resulting render and depth map have missing regions and Gaussian artifacts, which are rectified by an RGBD image-to-image diffusion model. This then acts as pseudo ground truth to spawn and update 3D Gaussians and satisfy the new view constraints. This process is repeated for several novel views spanning the 360°scene until the representation becomes multi-view consistent.

This section begins with an overview of our method in Sec. 2.1, detailing our approach for reconstructing a 3D scene from a sparse set of uncalibrated 2D images. In Sec. 2.1, we describe how we initialize a Gaussian point cloud using MASt3R and 3DGS to provide a coarse 3D representation. Sec. 2.2 introduces our RGBD

image-to-image generative model, which refines rendered novel views by correcting artifacts and filling in missing details. We propose a confidence measure (Sec. 2.3) based on cumulative transmittance and Gaussian density to guide the generative model toward unreliable regions. Sec. 2.4 outlines our synthetic dataset creation process for training with high-quality RGBD supervision, followed by depth-augmented autoencoder finetuning (Sec. 2.5) and UNet finetuning with synthesized data (Sec. 2.5.1) to improve generation quality. Finally, Secs. 2.5.2 and 2.6 detail the inference pipeline and 3D Gaussian optimization process, and Sec. 2.7 describes a test-time pose refinement step.

## 2.1 Algorithm Overview

**Problem Setup**  Given a set of $M$ images $\mathcal{I} = \{I_1, I_2, \ldots, I_M\}$ of an underlying 3D scene with unknown intrinsics and extrinsics, our goal is to reconstruct the 3D scene, estimate the camera poses of a monocular camera at the $M$ training views, and synthesize novel views at evaluation time given by $N$ unseen test images $\{I_{M+1}, I_{M+2}, \ldots, I_{M+N}\}$.

---

**Algorithm 1** Gaussian Scenes Training

---

**Require:** Sparse image set $\mathcal{I} = \{I_1, \ldots, I_M\}$, densification interval $k$, iterations $N$
 1: Initialize 3D Gaussian primitives $\mathcal{G}$ from MASt3R 3D point cloud $\mathbf{P}$
 2: Optimize $\mathcal{G}$ using 3DGS on input views $\mathcal{I}$ and estimated cameras $\pi_{train} = \{\pi_1, \ldots, \pi_M\}$ for $1k$ iterations
 3: $\hat{\mathcal{I}} \leftarrow \mathcal{I}$                            ▷ Initialize training images
 4: **for** $t = 1, \ldots, N$ **do**
 5:     **if** $t \mod k = 0$ **then**
 6:         Sample novel pose $\pi_{novel}$ from elliptical trajectory fitted to $\pi_{train}$
 7:         $(\hat{I}, \hat{D}) \leftarrow R_{\pi_{novel}}(\mathcal{G})$                  ▷ Render image and depth map
 8:         Compute confidence map $\mathcal{C}$ (Eq. 2)
 9:         Extract semantic context & geometry embeddings $\{c_{clip}, c_{geo}\}$          ▷ Sec 2.2.2
10:         $(I_{ref}, D_{ref}) \leftarrow \text{Refine}(\hat{I}, \hat{D}, \mathcal{C}, c_{clip}, c_{geo})$      ▷ Refine $\hat{I}$, $\hat{D}$ with generative priors (Sec 2.5.2)
11:         $\hat{\mathcal{I}} \leftarrow \hat{\mathcal{I}} \cup \{I_{ref}\}$             ▷ Synthesized novel view added as training sample
12:     **end if**
13:     Sample pose $\pi_i$ from $\pi_{train}$
14:     Render $(I_\pi, D_\pi) \leftarrow R_{\pi_i}(\mathcal{G})$
15:     Optimize 3D Gaussians $\mathcal{G}$                             ▷ Eq. 6 or 3DGS loss
16: **end for**

---

An overview of our method is given in Fig 3 and Alg 1. We initialize *GScenes* with an incomplete dense Gaussian point cloud reconstruction from sparse input images using MASt3R and *1k* iterations of 3DGS. Note that InstantSplat proposes the same framework for sparse-view reconstruction, but with a DUSt3R initialization. We choose MASt3R to initialize scene geometry instead due to its superior performance in the sparse-view setting. We use this incomplete scene representation as an implicit geometric prior and sample novel views along a smooth elliptical trajectory fitted to training views. An example trajectory is shown in Fig 4. We then use our RGBD

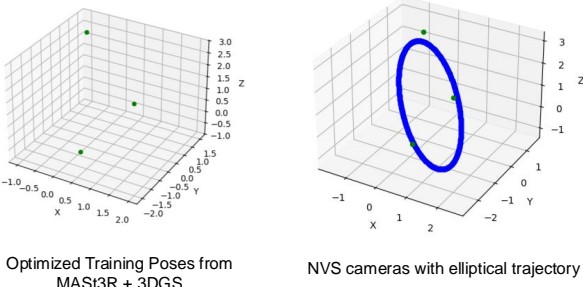

Optimized Training Poses from MASt3R + 3DGS      NVS cameras with elliptical trajectory

Figure 4: Camera Trajectory Visualization for Novel View Synthesis in pose-free sparse-view setting.

generative prior to synthesize plausible novel views. In addition to CLIP features of source images for context and plücker embeddings of source and target cameras for geometric conditioning, we devise a novel 3DGS confidence measure to effectively guide our generative model towards empty regions and potential artifacts in a novel view render. We then run *10k* iterations of 3DGS optimization, sampling a novel view before each densification step (Fu et al., 2023) to obtain our final scene representation.

**Gaussian Point Cloud Initialization**  The MASt3r initialization gives us a pixel-aligned dense-stereo point cloud $\mathbf{P} \in \mathbb{R}^{S \times 3}$, camera intrinsics $\{\mathbf{K_i} \in \mathbb{R}^{3 \times 3}\}_{i=1}^M$ and extrinsics $\{\mathbf{E_i} = [\mathbf{R_i}|\mathbf{T_i}]\}_{i=1}^M$ for our $M$ input images. Nevertheless, both $\mathbf{P}$ and the estimated poses demonstrate sub-optimal alignment compared to those generated by COLMAP from the dense observation dataset. We cannot also use COLMAP for camera or geometry estimation as the method fails when there is minimal overlap between training images, as in our setting. Consequently, similar to the approach in InstantSplat, we initialize 3D Gaussians at each location in the globally aligned point cloud $\mathbf{P}$. We then jointly optimize both the Gaussian attributes and camera parameters over the $1k$ iterations. The MASt3R point cloud is already quite dense ( 1.5-3M points), which alleviates the need for any form of Adaptive Density Control (Kerbl et al., 2023).

## 2.2 RGBD Generative Priors for Novel View Synthesis (NVS)

Reconstructing complete 3D scenes from sparse observations requires inferring content in unobserved regions—a fundamental challenge that geometric regularization and 3D priors alone cannot adequately address. We introduce a diffusion-based generative approach that leverages 2D image priors to synthesize plausible content.

### 2.2.1 Generative Model Architecture

Despite our initial geometric reconstruction, sparse input views inevitably result in regions with no Gaussian primitives ("0-Gaussians"), causing empty areas and artifacts in novel views. Unlike regularization-based methods that merely constrain optimization without generating content, our approach directly synthesizes missing scene details.

Our model comprises (1) A variational autoencoder (encoder $\mathcal{E}$, decoder $\mathcal{D}$) operating in a compressed latent space. (2) A UNet denoiser $\epsilon_\theta$ predicting noise in diffused latent $z_t$ (3) A multi-modal conditioning incorporating RGBD renders, confidence maps, semantic context, and geometric information.

The UNet $\epsilon_\theta$ receives four inputs: an artifact-laden RGBD image $\hat{I}$, a confidence map $\mathcal{C}$ identifying unreliable regions, CLIP features $c_{\text{clip}}$ of $\mathcal{I}$ providing semantic context, and camera encodings $c_{\text{geo}}$ establishing geometric relationships between views.

### 2.2.2 Multi-modal Conditioning

We initialize $\epsilon_\theta$ with pretrained Stable-Diffusion-2 model weights and expand the first convolutional layer to accept additional inputs by concatenating the noisy latent $z_t$, the encoded RGBD image $\mathcal{E}(\hat{I})$, the confidence-weighted encoded image $\mathcal{E}(\hat{I} \cdot \mathcal{C})$ and the downsampled confidence map $\hat{\mathcal{C}}$. To ensure view coherence and geometric consistency, we incorporate following conditioning signals:

**Semantic Context:** CLIP features $c_{\text{clip}} \in \mathbb{R}^{M \times d}$ from source images serve as semantic anchors, ensuring generated content remains consistent with observed scene elements.

**Geometric Information:** For each camera with center $\mathbf{o}$ and forward axis $\mathbf{d}$, we compute its plücker coordinates $\mathbf{r} = (\mathbf{d}, \mathbf{o} \times \mathbf{d}) \in \mathbb{R}^6$ and apply frequency encoding for obtaining higher-dimensional features:

$$\mathbf{r} \mapsto [\mathbf{r}, \sin(f_1 \pi \mathbf{r}), \cos(f_1 \pi \mathbf{r}), \cdots, \sin(f_K \pi \mathbf{r}), \cos(f_K \pi \mathbf{r})] \tag{1}$$

where $K = 6$ is the number of Fourier bands, and $f_k$ are equally spaced frequencies. This yields a 78-dimensional embedding for each camera ($c_{\text{geo}} \in \mathbb{R}^{(M+1) \times 78}$), capturing geometric relationships between viewpoints. Plücker coordinates were originally introduced by LFNs (Sitzmann et al., 2021) for per-pixel parameterization of a ray. We instead obtain a single representation per camera using extrinsic parameters $\mathbf{E_i}$ for obtaining $\mathbf{o}$ and $\mathbf{d}$.

### 2.2.3 Parameter-Efficient FiLM Conditioning

We employ Feature-wise Linear Modulation (FiLM) instead of cross-attention for incorporating context and geometry information, achieving both computational efficiency and strong performance. We process context and geometry embeddings through self-attention to capture inter-view relationships as: $c_{\text{attn}}^i = \text{SelfAttention}(c_i); \quad i \in \{\text{clip}, \text{geo}.\}$ We then generate scaling and shifting parameters via layer-specific

networks as: $\gamma^{(l)}, \beta^{(l)} = \text{FC}^{(l)}(c_{\text{attn}}^i)$. Finally, we modulate the UNet feature maps through element-wise operations as: $\mathbf{F}_{\text{mod}}^{(l)} = \gamma^{(l)} \cdot \mathbf{F}^{(l)} + \beta^{(l)}$.

Critically, we apply FiLM modulation only to down and mid blocks of the UNet—not up blocks—based on empirical evidence showing this selective application yields optimal results (Fig. 5).

Our FiLM-based approach requires only 8.14M parameters (7.38M for CLIP features, 758K for pose embeddings) compared to 29.8M for an equivalent cross-attention implementation—a 3× reduction while maintaining comparable quality, enabling more efficient training and faster inference during iterative reconstruction.

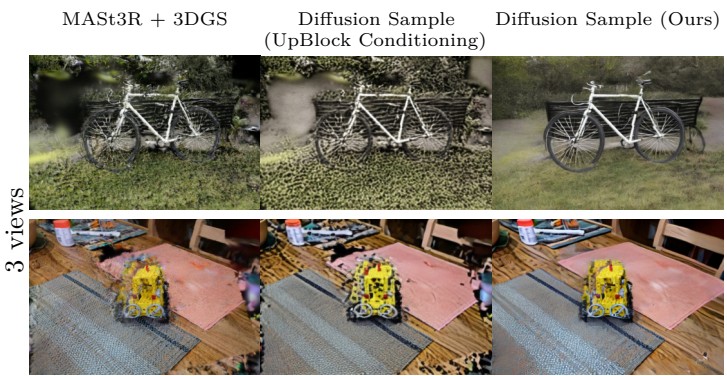

Figure 5: Incorporating context and geometry conditioning in the up blocks of the UNet negatively impacts latent and subsequent image reconstruction.

## 2.3 Pixel-Aligned Confidence Map

To guide our generative diffusion model in identifying problematic and artifact-ridden regions in novel views, we introduce a pixel-aligned confidence measure combining transmittance with Gaussian density:

$$\mathcal{C}_i = -\log(T_i + \epsilon) \times n_{\text{contrib}} \tag{2}$$

where $T_i = \prod_i (1 - \alpha_i)$ represents light transmission without Gaussian interaction, $n_{\text{contrib}}$ counts contributing 3D Gaussians, and $\epsilon > 0$ prevents logarithmic singularities. This formulation captures two complementary reliability signals: (1) low transmittance indicates significant Gaussian interactions, suggesting higher rendering confidence, and (2) consensus among multiple Gaussians validates pixel reliability through primitive agreement.

Unlike 3DGS-Enhancer (Liu et al., 2024), (one of the few methods that propose confidence measures for sparse-view scene reconstruction), which assumes that well-reconstructed areas contain small-scale Gaussians, our measure remains effective

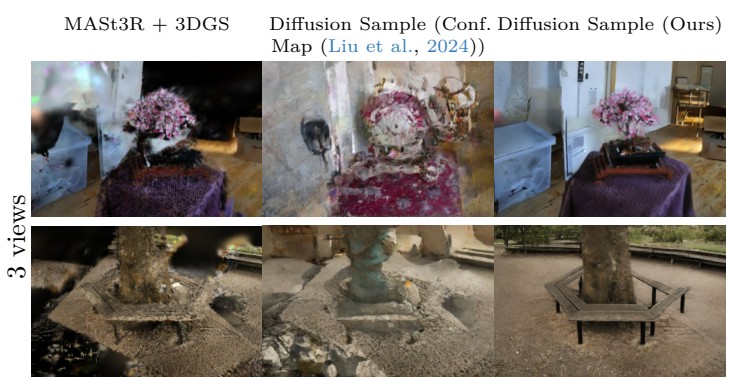

Figure 6: As an ablation for our confidence measure, we train a variant of our diffusion model using the confidence measure proposed in 3DGS Enhancer (Liu et al., 2024). Conditioning the UNet with this inaccurate confidence measure leads to implausible NVS.

for monotonous textures, where fine-grained Gaussian representation is unnecessary. Such regions are usually represented by a few high-opacity large-scale Gaussians (low $n_{contrib}$, but also low $T_i$) and hence not flagged as low-confidence regions by our confidence measure. Fig. 7 demonstrates how our approach accurately identifies both empty regions and reconstruction artifacts while avoiding false positives. The significance of this improved confidence measure is evident in Fig. 6, where models trained with previous confidence formulations produce implausible novel views due to misleading confidence signals.

## 2.4 RGBD Dataset Creation

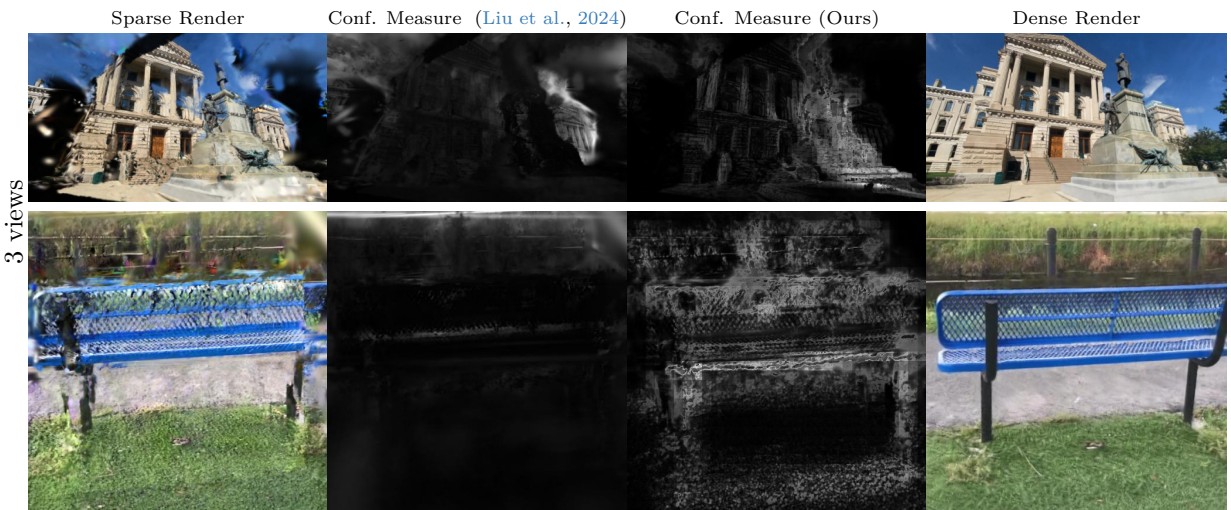

Figure 7: Confidence Measure comparison with 3DGS-Enhancer (Liu et al., 2024). Our confidence map accurately identifies artifacts and 0-Gaussian regions in the sparse-view (darker pixels) while Liu et al. (2024) incorrectly attributes high confidence to regions with overlap of small-scale Gaussians. NVS render from a densely fitted 3DGS representation is provided for reference.

For training the additional weights in $\epsilon_{\theta}$ for RGBD image-to-image diffusion, we rely on a set $\mathcal{X} = \{(I^i, \hat{I}^i, \mathcal{C}^i, c_{clip}^i, c_{geo}^i)_{i=1}^N\}$, each containing a clean RGBD image $I^i$, an RGBD image with artifacts $\hat{I}^i$ and the corresponding confidence map $\mathcal{C}^i$, CLIP features of source images $c_{clip}^i \in \mathcal{R}^{M \times 768}$, and plücker embeddings of source and target cameras $c_{geo}^i \in \mathcal{R}^{(M+1) \times 78}$, to "teach" the diffusion model how to inpaint missing details and detect Gaussian artifacts guided by the confidence map, context and geometry features and generate a clean version of the conditioning image. For this, we build a dataset generation pipeline comprising a high-quality 3DGS model fitted to dense views, a low-quality 3DGS model fitted to few views, and camera interpolation and perturbation modules to use supervision of the high-quality model at viewpoints beyond ground truth camera poses. For a given scene, we fit sparse models for $M \in \{3, 6, 9, 18\}$ number of views. We render $I^i$ using the high-quality model and $\hat{I}^i, \mathcal{C}^i$ using the low-quality model. We save the CLIP features $c_{clip}$ and plücker embeddings of the $M$ sparse views and 1 target view per sample for conditioning $\epsilon_{\theta}$. Our fine-tuning setup is illustrated in Fig 12 (Sec C).

## 2.5 Depth-Augmented Autoencoder Finetuning

For encoding and decoding RGBD images, we customize a Variational AutoEncoder by introducing additional channels in the first and last convolutional layers of the Stable Diffusion VAE. A similar approach was followed by Stan et al. (2023), where their KL-autencoder was finetuned with triplets containing RGB images, depth maps, and captions to train the weights in the new channels. However, the depth maps used for fine-tuning this VAE were estimated using MiDaS (Ranftl et al., 2020), which are usually blurry monocular depth estimates. As such, reconstructing RGBD images using this VAE produces depth maps with extreme blur - not ideal for a scene reconstruction problem. Hence, we further finetune this VAE with our syn-

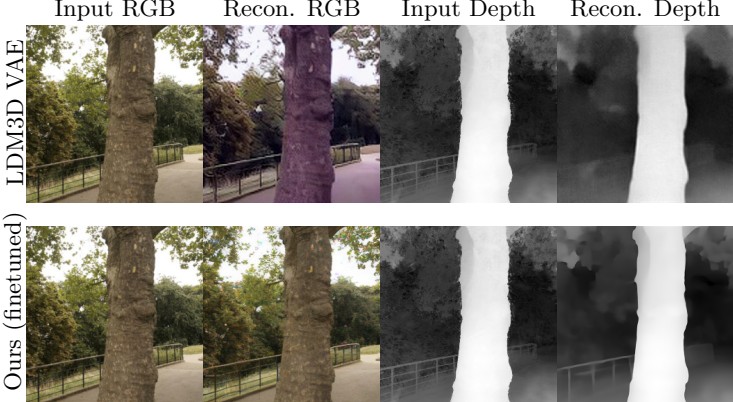

Figure 8: RGBD reconstruction comparison of our finetuned VAE with the LDM3D VAE (Stan et al., 2023). Unlike LDM3D, our VAE finetuned on a synthetic dataset preserves sharp details and edges of the input depth map while also preventing color artifacts in RGB.

thetic dataset, which contains depth maps rendered by the differentiable 3DGS rasterizer, giving accurate pixel depth with high-frequency details. Specifically, we use the following objective:

$$\mathcal{L}_{\text{autoencoder}} = \min_{\mathcal{E}, D} \max_{D_\psi} \big(\mathcal{L}_{\text{rec}}(x, D(\mathcal{E}(x))) - \mathcal{L}_{\text{adv}}(D(\mathcal{E}(x))) + \log(D_\psi(x)) + \mathcal{L}_{\text{reg}}(x; \mathcal{E}, D)\big) \tag{3}$$

where $\mathcal{L}_{rec}$ is a combination of L1, perceptual losses for the RGB channels, and Pearson Correlation Coefficient (PCC), TV regularization losses for the depth channels. $\mathcal{L}_{\text{adv}}$ is the adversarial loss, $D_\psi$ is a patch-based discriminator loss, and $\mathcal{L}_{\text{reg}}$ is the KL-regularisation loss. The incorporation of PCC and TV terms for the depth channels leads to better retention of high-frequency details in the reconstructed depth map, as observed in Fig 8. We finetune this VAE on a subset of our dataset for 5000 training steps with batch size 16 and learning rate 1e-05.

### 2.5.1 UNet Finetuning

With our finetuned autoencoder, we next train the UNet with the frozen VAE on $\mathcal{X}$ with this objective:

$$\mathcal{L}_{diff} = \mathbb{E}_{i \sim \mathcal{U}(N), \epsilon \sim \mathcal{N}(\mathbf{0}, \mathbf{I}), t} \left[ \| \epsilon_t - \epsilon_{\boldsymbol{\theta}}(\mathbf{z}_t^i; t, \mathcal{E}(\hat{I}), \hat{\mathcal{C}}, \mathcal{E}(\hat{I} \cdot \mathcal{C}), c_{attn}^{clip}, c_{attn}^{geo}) \|_2^2 \right] \tag{4}$$

### 2.5.2 RGBD Novel View Synthesis

At inference time, given a render and depth map with artifacts, and confidence map, CLIP features and camera embeddings for conditioning, the finetuned UNet $\epsilon_{\boldsymbol{\theta}}$ learns to predict the noise in latent $\mathbf{z}_t$ according to $t \sim \mathcal{U}[t_{min}, t_{max}]$ as:

$$\begin{aligned} \hat{\epsilon}_t = & \, \epsilon_{\boldsymbol{\theta}}(\mathbf{z}_t; t, \varnothing, \varnothing, \varnothing, c_{attn}^{clip}, c_{attn}^{geo}) \\ & + s_I(\epsilon_{\boldsymbol{\theta}}(\mathbf{z}_t; t, \mathcal{E}(\hat{I}), \hat{\mathcal{C}}, \mathcal{E}(\hat{I} \cdot \mathcal{C}), c_{attn}^{clip}, c_{attn}^{geo}) - \epsilon_{\boldsymbol{\theta}}(\mathbf{z}_t; t, \varnothing, \hat{\mathcal{C}}, \varnothing, c_{attn}^{clip}, c_{attn}^{geo})) \\ & + s_C(\epsilon_{\boldsymbol{\theta}}(\mathbf{z}_t; t, \varnothing, \hat{\mathcal{C}}, \varnothing, c_{attn}^{clip}, c_{attn}^{geo}) - \epsilon_{\boldsymbol{\theta}}(\mathbf{z}_t; t, \varnothing, \varnothing, \varnothing, c_{attn}^{clip}, c_{attn}^{geo})) \end{aligned} \tag{5}$$

where $s_I$ and $s_C$ are the RGBD image and confidence map guidance scales, dictating how strongly the final multistep reconstruction agrees with the RGBD render $\hat{I}$ and the confidence map $\mathcal{C}$, respectively. After $k = 20$ DDIM (Song et al., 2021) sampling steps, we obtain our final RGBD render by decoding the denoised latent as $x_\pi = [I_\pi, D_\pi] = \mathcal{D}(z_0)$.

### 2.6 Scene Reconstruction with Generative Priors

Our generative diffusion model provides a generative prior to infer plausible detail in unobserved regions. Despite view conditioning using pose embeddings, the generated images at novel poses lack complete 3D consistency. For this, we devise an iterative strategy where we first sample novel views along an elliptical trajectory fitted to the training views. We initialize the Gaussian optimization with the set of Gaussians $\mathcal{G}$ fitted to the training views (Sec 2.1). Each novel view is added to the training stack at the beginning of every densification step to encourage the optimization to adjust to the distilled scene priors. At every iteration, we sample either an observed or unobserved viewpoint from the current training stack. We bring back Adaptive Density Control to encourage densification of Gaussians in 0-Gaussian regions. We employ the 3DGS objective for the training views. For novel views, we employ the SparseFusion (Zhou and Tulsiani, 2023) objective in the RGB space and a PCC loss for the rendered and denoised depths.

$$\mathcal{L}_{sample}(\psi) = \mathbb{E}_{\pi, t} \left[ w(t)(\|I_\pi - \hat{I}_\pi\|_1 + \mathcal{L}_p(I_\pi, \hat{I}_\pi)) \right] + w_d \cdot PCC(D_\pi, \hat{D}_\pi) \tag{6}$$

where $\mathcal{L}_p$ is the perceptual loss (Zhang et al., 2018), $w(t)$ a noise-dependent weighting function, $I_\pi, D_\pi$ are the rendered image and depth at novel viewpoint $\pi$, and $\hat{I}_\pi, \hat{D}_\pi$ are their rectified versions obtained with our diffusion prior. The PCC loss is defined as $PCC(D_\pi, \hat{D}_\pi) = 1 - \frac{\text{Cov}(D_\pi, \hat{D}_\pi)}{\sigma_{D_\pi} \sigma_{\hat{D}_\pi}}$.

## 2.7 Test-time Pose Alignment

*GScenes* reconstructs a plausible 3D scene from pose-free source images. However, reconstruction with few views is inherently ambiguous as several solutions can satisfy the train view constraints. Hence, the reconstructed scene would probably be quite different from the actual scene from which $M$ views were sampled. Hence, for a given set of test views, following prior work (Fan et al., 2024; Jiang et al., 2024), we freeze the Gaussian attributes and optimize the camera pose for each target view by minimizing a photometric loss between the rendered image and test view. Following this alignment step performed for 500 iterations per test image, we evaluate the NVS quality.

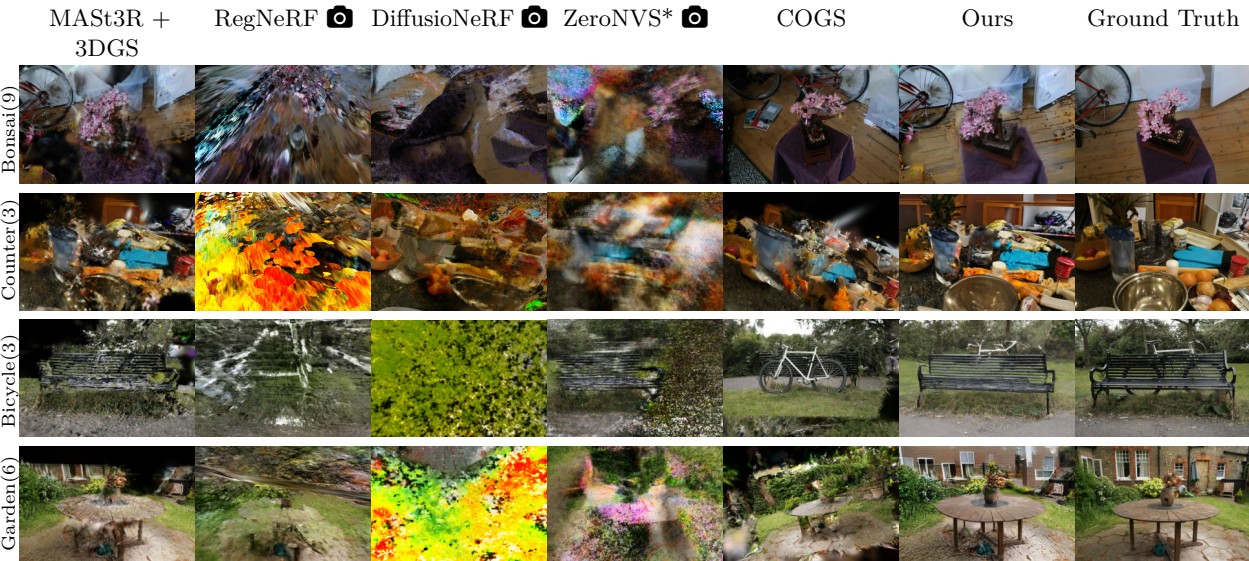

Figure 9: **Qualitative comparison** of GScenes with few-view methods on the **MipNeRF360** dataset. Our approach consistently fairs better in recovering image structure from foggy geometry, where baselines typically struggle with "floaters" and color artifacts.

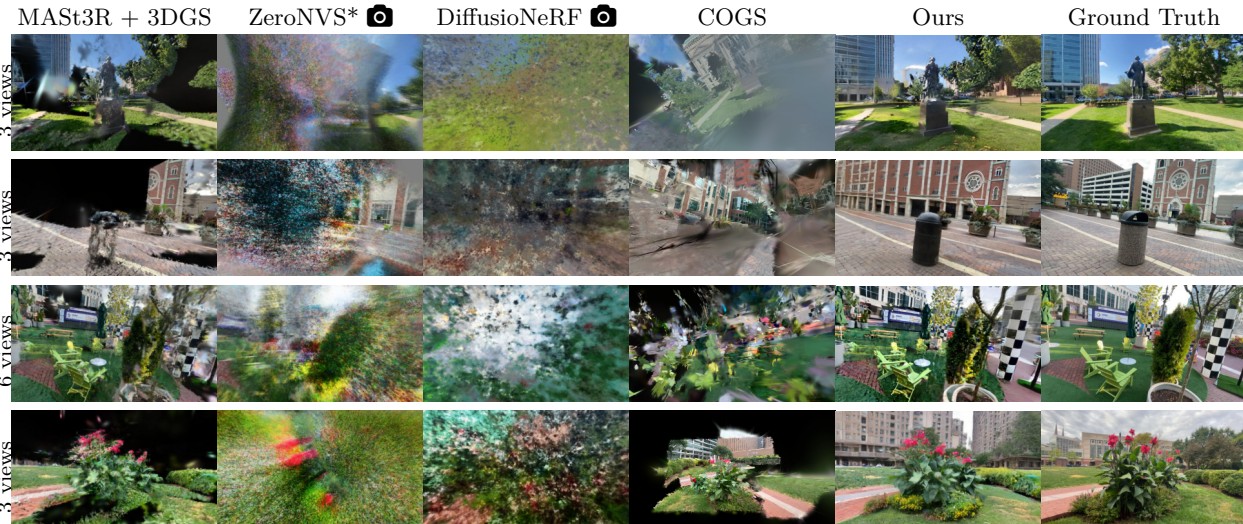

Figure 10: **Qualitative comparison** of *GScenes* with few-view methods on the **DL3DV benchmark**. Our method achieves plausible reconstruction in unobserved areas of complex scenes where even posed reconstruction techniques struggle.

# 3 Experiments

We compare *GScenes* with state-of-the-art pose-free and posed sparse-view reconstruction methods in Fig 9, 10 and Table 1, 2. We also ablate the different components and design choices of our diffusion model.

## 3.1 Experimental Setup

**Evaluation Dataset**   We evaluate GScenes on the 9 scenes of the MipNeRF360 dataset (Barron et al., 2022), and 15 scenes (out of 140) of the DL3DV-10K benchmark dataset. For MipNeRF360, We pick the $M$-view splits as proposed by ReconFusion and CAT3D and evaluate all baselines on the official test views where every $8^{th}$ image is held out for testing. For DL3DV-10K scenes, we create $M$-view splits using a greedy view-selection heuristic for maximizing scene coverage given a set of dense training views, similar to the heuristic proposed in Wu et al. (2024). For test views, we hold out every $8^{th}$ image as in MipNeRF360. Additionally, we pick the *plant* scene of CO3D for qualitative comparison with ReconFusion and CAT3D.

**Fine-tuning Dataset**   We fine-tune our diffusion model on a mix of 1043 scenes encompassing Tanks and Temples (Knapitsch et al., 2017), CO3D (Reizenstein et al., 2021), Deep Blending (Hedman et al., 2018), and the $1k$ subset of DL3DV-10K (Ling et al., 2024) to obtain a total of $171,461$ data samples. We first train 3DGS on sparse and dense subsets of each scene for $M \in \{3, 6, 9, 18\}$. For $M > 18$, novel view renders and depth maps mostly show Gaussian blur as artifacts. Finetuning this model takes about 4-days on a single A6000 GPU.

**Metrics**   Our quantitative metrics are used to evaluate two tasks - quality of novel views post reconstruction and camera pose estimation. For the former, we compute 3 groups of metrics - FID (Heusel et al., 2017) and KID (Bińkowski et al., 2018) due to the generative nature of our approach, perceptual metrics LPIPS (Zhang et al., 2018) and DISTS (Ding et al., 2020) to measure similarity in image structure and texture in the feature space, and pixel-aligned metrics PSNR and SSIM. However, PSNR and SSIM are not suitable evaluators of generative techniques (Chan et al., 2023; Sargent et al., 2024) as they favor pixel-aligned blurry estimates over high-frequency details.

**Baselines**   We compare our approach against 8 baselines, but our main comparison is with 2 pose-free methods - COGS and our 3D reconstruction engine - MASt3R + 3DGS. The other 6 methods are posed and included for a stronger comparison. Out of these, FreeNeRF, RegNeRF, and DiffusioNeRF are sparse-view regularization methods based on NeRFs. ZeroNVS reconstructs a complete 3D scene from a single image using a novel camera normalization scheme and anchored SDS loss. We use the ZeroNVS* baseline introduced in ReconFusion, designed to adapt ZeroNVS to multi-view inputs. Further, we also provide the reported quantitative results from ReconFusion and CAT3D, which are closed source and unverifiable, to show that our method gets close to their performance despite using a fraction of their compute. We also pick the relevant scenes and test views from the 2 papers for qualitative comparison.

## 3.2 Implementation Details

*GScenes* is implemented in PyTorch 2.3.1 on single A5000/A6000 GPUs. Images and depth maps are rendered at 400-600 pixels to align with Stable Diffusion's resolution. The diffusion model is finetuned for $100k$ iterations (batch size 16, learning rate 1e-4) with conditioning element dropout probability of 0.05 for CFG.

Following InstantSplat, we fit 3D Gaussians to sparse inputs and MASt3r point clouds for $1k$ iterations to obtain $\mathcal{G}$. We use classifier-free guidance scales $s_I = s_C = 3.0$ and sample with $k = 20$ DDIM steps. We linearly decay $w_d$ from 1 to 0.01 and $\mathcal{L}_{sample}$ weight from 1 to 0.1 over $10k$ iterations. *GScenes* completes full 3D reconstruction in approximately 5 minutes on a single A6000 GPU.

## 3.3 Comparative Results

We report qualitative and quantitative comparisons of *GScenes* against all related baselines in Fig 9, 10 and Tables 1 and 2. Additional qualitative results are provided in Fig 14 and 15 (Sec E). Out of the 8 baselines, only *MASt3R + 3DGS* and *COGS* are pose-free techniques, but *COGS* relies on ground truth camera intrinsics while *GScenes* and *MASt3R + 3DGS* do not. In classical metrics (Tab 1), our baseline *MASt3R +*

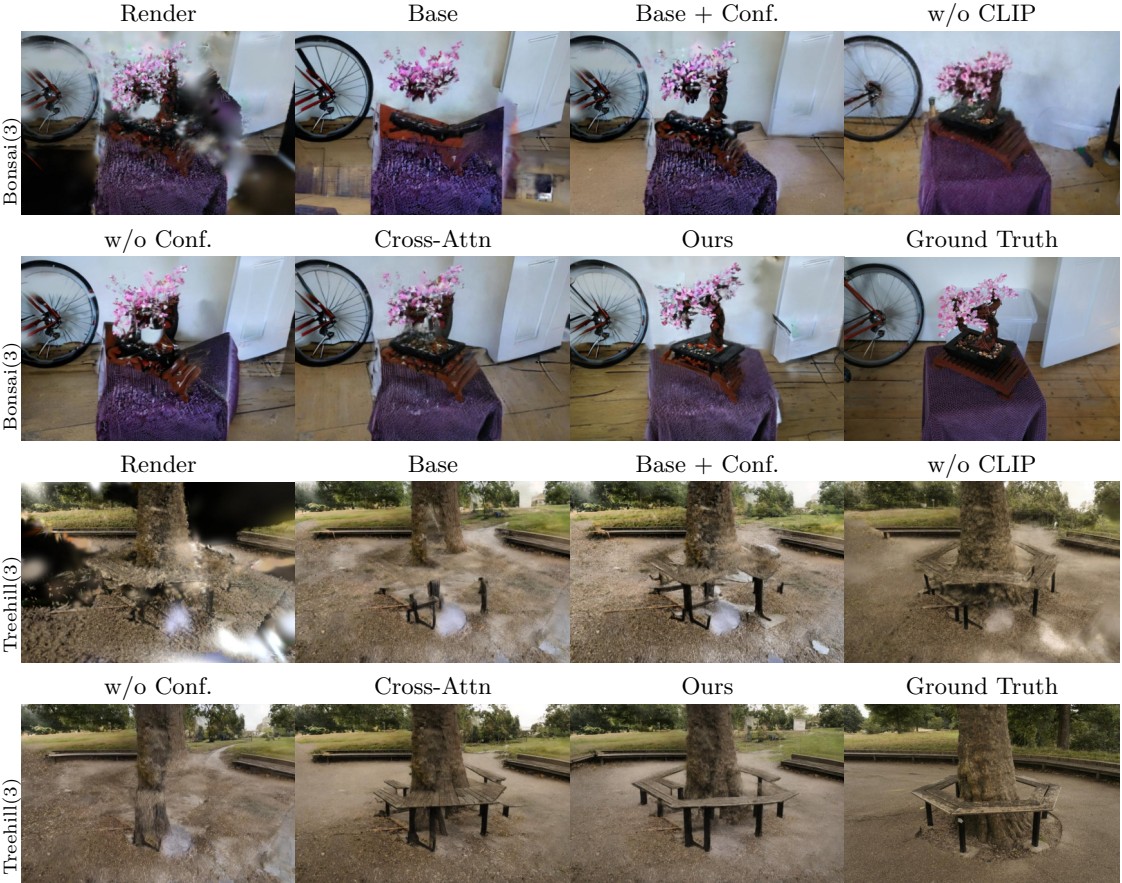

Figure 11: Ablation Study with the 3-view splits of *Bonsai* and *Treehill* scenes from MipNeRF360. Images show samples from different variants of our diffusion model based on conditioning.

Table 1: **Quantitative comparison** with state-of-the-art sparse-view reconstruction techniques on classical metrics. Red, orange, and yellow indicate best, second-best, and third-best performing methods, respectively. 📷 indicates posed methods.

| | | PSNR ↑ | | | SSIM ↑ | | | LPIPS ↓ | | |
|---|---|---|---|---|---|---|---|---|---|---|
| | Method | 3-view | 6-view | 9-view | 3-view | 6-view | 9-view | 3-view | 6-view | 9-view |
| MipNeRF360 | FreeNeRF 📷 | 11.886 | 12.874 | 13.673 | 0.180 | 0.218 | 0.236 | 0.675 | 0.654 | 0.638 |
| | RegNeRF 📷 | 12.294 | 13.204 | 13.796 | 0.182 | 0.207 | 0.217 | 0.668 | 0.656 | 0.625 |
| | DiffusioNeRF 📷 | 9.996 | 12.025 | 12.567 | 0.159 | 0.211 | 0.213 | 0.715 | 0.648 | 0.659 |
| | ZeroNVS* 📷 | 11.991 | 11.817 | 11.729 | 0.142 | 0.129 | 0.128 | 0.710 | 0.705 | 0.694 |
| | ReconFusion 📷 | 15.50 | 16.93 | 18.19 | 0.358 | 0.401 | 0.432 | 0.585 | 0.544 | 0.511 |
| | CAT3D 📷 | 16.62 | 17.72 | 18.67 | 0.377 | 0.425 | 0.460 | 0.515 | 0.482 | 0.460 |
| | MASt3R + 3DGS | 12.585 | 14.285 | 15.042 | 0.231 | 0.279 | 0.310 | 0.593 | 0.550 | 0.531 |
| | COGS | 11.814 | 12.095 | 12.555 | 0.183 | 0.210 | 0.228 | 0.619 | 0.630 | 0.605 |
| | *GScenes* | 13.809 | 14.507 | 14.831 | 0.265 | 0.275 | 0.282 | 0.547 | 0.530 | 0.521 |
| DL3DV | DiffusioNeRF 📷 | 11.715 | 13.245 | 14.504 | 0.245 | 0.317 | 0.376 | 0.606 | 0.544 | 0.500 |
| | ZeroNVS* 📷 | 12.145 | 12.560 | 12.774 | 0.202 | 0.206 | 0.209 | 0.679 | 0.658 | 0.648 |
| | MASt3R + 3DGS | 13.217 | 16.417 | 17.840 | 0.371 | 0.476 | 0.531 | 0.443 | 0.359 | 0.326 |
| | COGS | 11.057 | 12.350 | 13.300 | 0.241 | 0.282 | 0.318 | 0.598 | 0.564 | 0.539 |
| | *GScenes* | 14.751 | 15.761 | 16.199 | 0.340 | 0.372 | 0.394 | 0.410 | 0.373 | 0.360 |

Table 2: **Quantitative comparison** with few-view reconstruction techniques on metrics suited for generative reconstruction. Red, orange, and yellow indicate best, second-best, and third-best performing methods.

| | Method | FID ↓ | | | KID ↓ | | | DISTS ↓ | | |
|---|---|---|---|---|---|---|---|---|---|---|
| | | 3-view | 6-view | 9-view | 3-view | 6-view | 9-view | 3-view | 6-view | 9-view |
| MipNeRF360 | FreeNeRF 📷 | 354.244 | 353.480 | 346.874 | 0.261 | 0.271 | 0.282 | 0.388 | 0.369 | 0.365 |
| | RegNeRF 📷 | 358.855 | 360.380 | 338.183 | 0.281 | 0.301 | 0.264 | 0.407 | 0.405 | 0.377 |
| | DiffusioNeRF 📷 | 370.346 | 347.522 | 342.373 | 0.293 | 0.267 | 0.257 | 0.441 | 0.386 | 0.402 |
| | ZeroNVS* 📷 | 356.395 | 350.920 | 343.930 | 0.283 | 0.296 | 0.300 | 0.433 | 0.426 | 0.413 |
| | MASt3R + 3DGS | 294.819 | 252.830 | 231.169 | 0.210 | 0.164 | 0.133 | 0.303 | 0.273 | 0.264 |
| | COGS | 251.694 | 284.829 | 270.118 | 0.158 | 0.183 | 0.159 | 0.297 | 0.313 | 0.303 |
| | *GScenes* | 163.747 | 156.368 | 160.919 | 0.053 | 0.052 | 0.055 | 0.234 | 0.227 | 0.230 |
| DL3DV | DiffusioNeRF 📷 | 251.341 | 210.596 | 199.629 | 0.179 | 0.154 | 0.147 | 0.352 | 0.314 | 0.300 |
| | ZeroNVS* 📷 | 267.407 | 243.095 | 233.998 | 0.214 | 0.200 | 0.199 | 0.372 | 0.346 | 0.330 |
| | MASt3R + 3DGS | 174.665 | 136.013 | 118.675 | 0.117 | 0.077 | 0.062 | 0.247 | 0.204 | 0.187 |
| | COGS | 249.530 | 224.605 | 207.782 | 0.198 | 0.159 | 0.128 | 0.308 | 0.295 | 0.283 |
| | *GScenes* | 107.750 | 107.329 | 105.871 | 0.036 | 0.046 | 0.046 | 0.193 | 0.190 | 0.189 |

*3DGS* shows better values than *GScenes*, as these metrics are usually inadequate for evaluating generative NVS techniques. Nevertheless, we include them for completeness and observe that either our method or the baseline is the best pose-free, open-source method. CAT3D and Reconfusion appear to do better in a posed setting, but we cannot confirm these values nor do a fair comparison in a pose-free setting as they are closed-source. For metrics tailored for generative reconstruction (Tab 2), our method comprehensively outperforms the baseline as well as all related methods for both MipNeRF360 and DL3DV-Benchmark datasets. Due to the unavailability of open-source code, evaluating *ReconFusion* and *CAT3D* on these measures is unfortunately not possible. However, we do provide a qualitative comparison with both methods in Sec D. Qualitative comparisons with other open-source posed and pose-free methods clearly show that *GScenes* is far more adept at generating plausible scene content from partial observations.

### 3.4 Ablation Studies

In Fig 11 and Tab 3 (Sec F), we thoroughly ablate different components of our diffusion model. We pick the *bonsai* and *treehill* scenes and their 3-view splits for this experiment. The $1^{st}$ out of 8 columns shows the initial novel-view render obtained from the 3D reconstruction pipeline (MASt3r + 3DGS). The *Base* variant only performs image-to-image diffusion with no other conditioning, and this already provides a strong baseline for inpainting and artifact elimination in novel-view renders. Without CLIP context guidance, the model fails to adhere to the semantics of the input images when inpainting missing details. Without our confidence measure, the model typically fails to differentiate between image structures and artifacts, often producing implausible images despite context and geometry conditioning. Additionally, we train a *Cross-Attn* variant where context and geometry conditionings are incorporated using cross-attention instead of FiLM modulation layers in the down and middle blocks of the UNet. Our method achieves similar quality and FID/KID/DISTS scores with more than **3x** fewer additional trainable parameters. Similar to Tab 1, *GScenes* performs a little worse on PSNR, SSIM, and LPIPS compared to the baseline, as these metrics penalize the generation of plausible scene details, slightly dissimilar from the original scene. Nevertheless, the qualitative results clearly show the benefits of our proposed context, geometry, and confidence map conditioning.

## 4 Conclusion

In this work, we present *GScenes* where we integrate an image-to-image RGBD diffusion model with a pose-free reconstruction pipeline in MASt3R to reconstruct a 360°3D scene from a few uncalibrated 2D images. We introduce context and geometry conditioning through FiLM layers, achieving similar performance as a cross-attention variant. We also introduce a pixel-aligned confidence measure to further guide the diffusion model in uncertain regions with missing details and artifacts. Our experiments show that *GScenes* outperforms existing pose-free reconstruction techniques in scene reconstruction and performs competitively with state-of-the-art posed sparse-view reconstruction methods.

## 5    Acknowledgement

This work is supported by Army Research Laboratory award W911NF2320008 and ONR with N00014-21-1-2812.

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

# A    Problem Introduction and Discussion

Obtaining high-quality 3D reconstructions or novel views from a sparse set of images has been a long-standing goal in computer vision. Recent methods for sparse-view reconstruction often employ generative, geometric, or semantic priors to stabilize the optimization of NeRFs (Mildenhall et al., 2020) or Gaussian splats (Kerbl et al., 2023) in highly under-constrained scenarios. However, they typically assume access to accurate intrinsic and extrinsic parameters, often derived from dense observations. This reliance on ground-truth poses is a restrictive assumption, making these methods impractical for real-world applications. Moreover, pose estimation from sparse views is error-prone; both traditional Structure from Motion approaches and recent 3D foundational models (Wang et al., 2024; Leroy et al., 2024) struggle with insufficient matching features between image pairs. In response, recent pose-free approaches using 3D Gaussian splatting (3DGS) integrate monocular depth estimation (Ranftl et al., 2020), 2D semantic segmentation (Kirillov et al., 2023), or 3D foundational priors (Wang et al., 2024), optimizing 3D Gaussians and camera poses together during training. However, these methods are typically designed for scenes with high view overlap, and they often fail to reconstruct complex, large-scale 360° scenes with sparse coverage. Additionally, despite extensive regularization to prevent overfitting, the limited observations impede coherent synthesis of unobserved regions. This challenge underlines the need for additional generative regularization to enable accurate extrapolation and complete scene reconstruction.

In the absence of poses estimated from dense observations, we instead rely on recent 3D foundational priors (Leroy et al., 2024) for scene initialization from sparse views and encode its estimated cameras for conditioning a Stable Diffusion UNet (Rombach et al., 2022) during both training and evaluation. We also augment the UNet with additional channels for context, 3DGS render, depth maps with Gaussian artifacts, and a pixel-aligned confidence map capturing missing regions and reconstruction artifacts in the RGBD image. During inference, the model predicts a clean, inpainted version of the conditioning artifact image and an aligned depth map. This formulation prevents the requirement of accurate ground truth poses for pose conditioning of a multiview diffusion model. This also alleviates dependency on large-scale 3D datasets which are usually synthetic and low quality compared to real-world scenes.

We present *GScenes*, an efficient method that uses 3D foundational (Leroy et al., 2024) and RGBD diffusion priors for pose-free sparse-view reconstruction of complex 360° scenes. Our method leverages stronger priors than simple regularizers while not relying on million-scale multi-view data or huge compute resources to train a 3D-aware diffusion model. We also ablate all conditioning elements in our diffusion model and identify which conditioning features contribute the most towards coherent NVS from sparse views.

# B    Related Work

Reconstructing 3D scenes from limited observations requires generative priors or, more specifically, inpainting missing details in unseen regions in 3D and removing artifacts introduced through observing scene areas from a few observations. Our work builds on recent developments in 2D diffusion priors for 3D reconstruction (Paul et al., 2024), where knowledge learned from abundant 2D datasets is lifted to 3D for refining novel views rendered by sparse 3D models. Next, we discuss how our work is related to the current line of research.

**Regularization Techniques**    Both NeRF and 3DGS rely on hundreds of scene captures for photorealistic novel view synthesis. When the input set becomes sparse, the problem becomes ill-posed, as several simultaneous 3D representations can agree with the training set. Regularization techniques are amongst the earliest techniques to address this limitation. Typical methods leverage depth from Structure-from-Motion (SfM)(Deng et al., 2022; Roessle et al., 2022), monocular estimation(Li et al., 2024; Xiong et al., 2023; Zhu et al., 2023; Chung et al., 2023), or RGB-D sensors (Wang et al., 2023a). DietNeRF (Jain et al., 2021) uses a semantic consistency loss based on CLIP (Radford et al., 2021) features, while FreeNeRF (Yang et al., 2023) regularizes the frequency range of NeRF inputs. RegNeRF (Niemeyer et al., 2022) and DiffusioNeRF (Wynn and Turmukhambetov, 2023) maximize the likelihoods of rendered patches using normalizing flows or diffusion models, respectively. However, such techniques usually fail under extreme sparsity like 3, 6, or 9 input images

for a 360° scene due to weaker priors. Generative priors can be viewed as a stronger form of regularization as they provide extrapolation capabilities for inferring details in unknown parts of a scene.

**Generalizable Reconstruction**   When only a few or a single view is available, regularization techniques are often insufficient to resolve reconstruction ambiguities. To address this, recent research focuses on training priors for novel view synthesis across multiple scenes. pixelNeRF (Yu et al., 2021) uses pixel-aligned CNN features as conditioning for a shared NeRF MLP, while other approaches (Trevithick and Yang, 2021; Chen et al., 2021; Henzler et al., 2021; Lin et al., 2023b; Szymanowicz et al., 2023) condition NeRF on 2D or fused 3D features. Further priors have been learned on triplanes (Irshad et al., 2023), voxel grids (Guo et al., 2022), neural points (Wewer et al., 2023), and IB-planes Anciukevičius et al. (2024). Leveraging 3D Gaussian Splatting (Kerbl et al., 2023), methods like pixelSplat (Charatan et al., 2024) and MVSplat (Chen et al., 2024) achieve state-of-the-art performance in stereo view interpolation. However, these regression-based techniques infer blurry novel views in case of high uncertainty.

**Generative Priors for NVS**   For ambiguous novel views, predicting expectations over all reconstructions may be unreliable. Consequently, regression approaches fall short, whereas generative methods attempt to sample from a multi-modal distribution.

While diffusion models have been applied directly on 3D representations like triplanes (Shue et al., 2023; Chen et al., 2023a), voxel grids (Müller et al., 2023), or (neural) point clouds (Zhou et al., 2021; Melas-Kyriazi et al., 2023; Schröppel et al., 2024), 3D data is scarce. Given the success of large-scale diffusion models for image synthesis, there is a great research interest in leveraging them as priors for 3D reconstruction and generation. DreamFusion (Poole et al., 2023) and follow-ups (Wang et al., 2023b; Lin et al., 2023a; Chen et al., 2023b; Deng et al., 2023; Tang et al., 2024) employ score distillation sampling (SDS) to iteratively maximize the likelihood of radiance field renderings under a conditional 2D diffusion prior. For sparse-view reconstruction, existing approaches incorporate view-conditioning via epipolar feature transform (Zhou and Tulsiani, 2023), cross-attention to encoded relative poses (Liu et al., 2023; Sargent et al., 2024), or pixelNeRF (Yu et al., 2021) feature renderings (Wu et al., 2024). However, this fine-tuning is expensive and requires large-scale multi-view data, which we circumvent with *GScenes*.

**Pose-Free 3D Reconstruction**   For building a generalizable sparse-view reconstruction method, the assumption of camera poses during inference limits applications to real-world scenarios where usually only an uncalibrated set of 2D images are available with no known camera extrinsics or intrinsics. Several recent works (Jiang et al., 2024; Fan et al., 2024) have attempted to solve reconstruction in a pose-free setting by jointly optimizing poses and NeRF or 3D Gaussian parameters during scene optimization. These methods typically outperform previous techniques (Chen and Lee, 2023; Lin et al., 2021; Bian et al., 2023; Fu et al., 2023) where reconstruction is done in two stages - first estimating poses and then optimizing the 3D representation. However, errors in the initial pose estimation harm subsequent scene optimization, resulting in inferior NVS quality. In our work, we use the MASt3R pipeline with 3DGS for predicting 3D Gaussians and camera parameters in a global coordinate system from a set of unposed 2D images.

Our work is most closely related to $Sp^2360$ (Paul et al., 2024) where an instruction-following RGB diffusion model is finetuned for the task of rectifying novel views rendered by 3DGS fitted to sparse observations. We extend the problem setting to the more challenging pose-free scenario and jointly model RGB and depth to aid optimization of 3D Gaussians during the distillation phase. Additionally, we introduce CLIP context and pose conditioning through FiLM layers and a pixel-aligned confidence measure for more accurate novel view synthesis.

## C   RGBD Dataset Creation

Our fine-tuning setup for the RGBD diffusion model is illustrated in Fig 12.

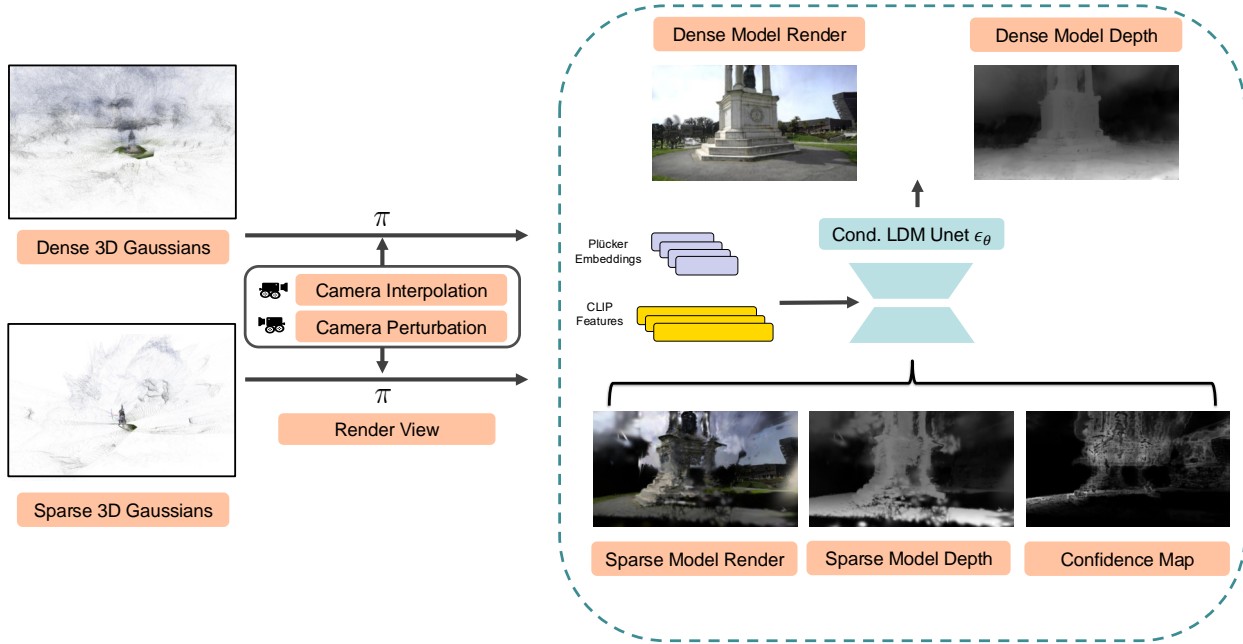

Figure 12: **Training our RGBD diffusion model.** Pairs of RGBD clean images and images with artifacts are obtained from 3DGS fitted to sparse and dense observations, respectively, across 1043 scenes. CLIP features provide semantic scene context, plücker embeddings of source and target cameras provide geometry information, and a confidence map additionally detects empty regions and artifacts in the artifact image. The Stable Diffusion UNet (Rombach et al., 2022) is then fine-tuned with a dataset of 171, 461 samples.

## D    Qualitative Comparison with ReconFusion / CAT3D

In Fig 13, we provide an additional qualitative comparison with ReconFusion and CAT3D. Their code, data, models, or training reproduction details are not available, and hence, we can not perform any evaluation across any of the test scenes in their paper. Despite this issue, from the figures in the 2 papers, we pick the relevant test views for the *treehill*, *flowers*, *bicycle* scenes in MipNeRF360, and the *plant* scene from CO3Dv2 to show how *GScenes* compares with their reconstruction. We use the same training views as open-sourced in their data splits. Despite being a pose-free pipeline using weaker generative priors, we observe that *GScenes* compares competitively with both methods.

## E    Additional Qualitative Results

Additional qualitative comparisons on MipNeRF360 and DL3DV-benchmark datasets are provided in Fig 14 and  15.

## F    Ablation Studies

We report the quantitative performance of our method with different variants of the proposed diffusion model in Tab 3.

## G    Evaluating DiffusioNeRF in a pose-free setting

Since the performance of ReconFusion and CAT3D in a pose-free setting cannot be ascertained due to a lack of open-source code or models, we instead evaluate DiffusioNeRF, an open-source posed reconstruction technique, on 3 scenes from the DL3DV-benchmark in the same pose-free evaluation protocol as COGS, MASt3R+3DGS, and *GScenes*. We use noisy poses estimated by MASt3R from sparse inputs as training

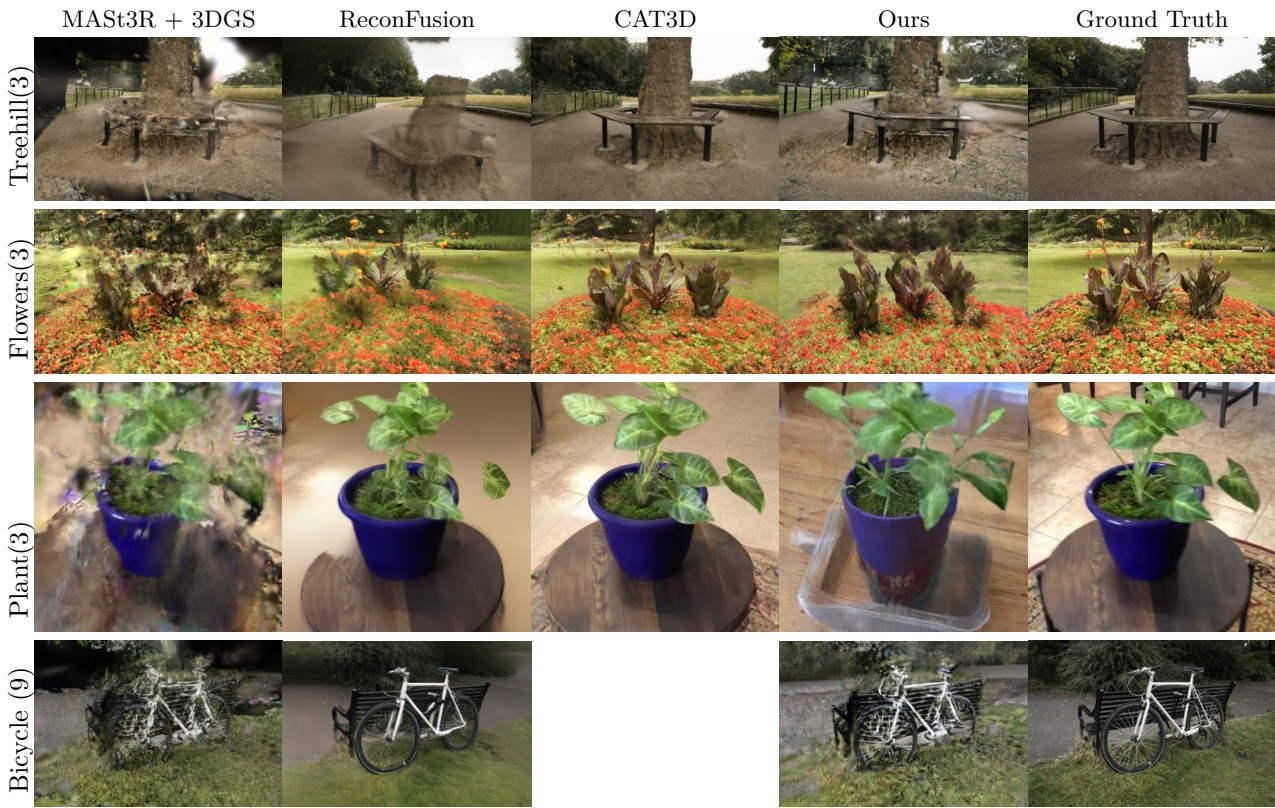

Figure 13: **Qualitative comparison** of *GScenes* with ReconFusion and CAT3D (posed techniques). Despite being a pose-free pipeline built with weaker diffusion priors, our method achieves better (*flowers*, *treehill*) or similar (*plant*) NVS quality compared to ReconFusion on three out of 4 examples above. We are slightly worse compared to closed-source CAT3D (although their results cannot be validated), which uses a closed-source, stronger video diffusion prior. No image available for CAT3D in the last row, hence kept blank.

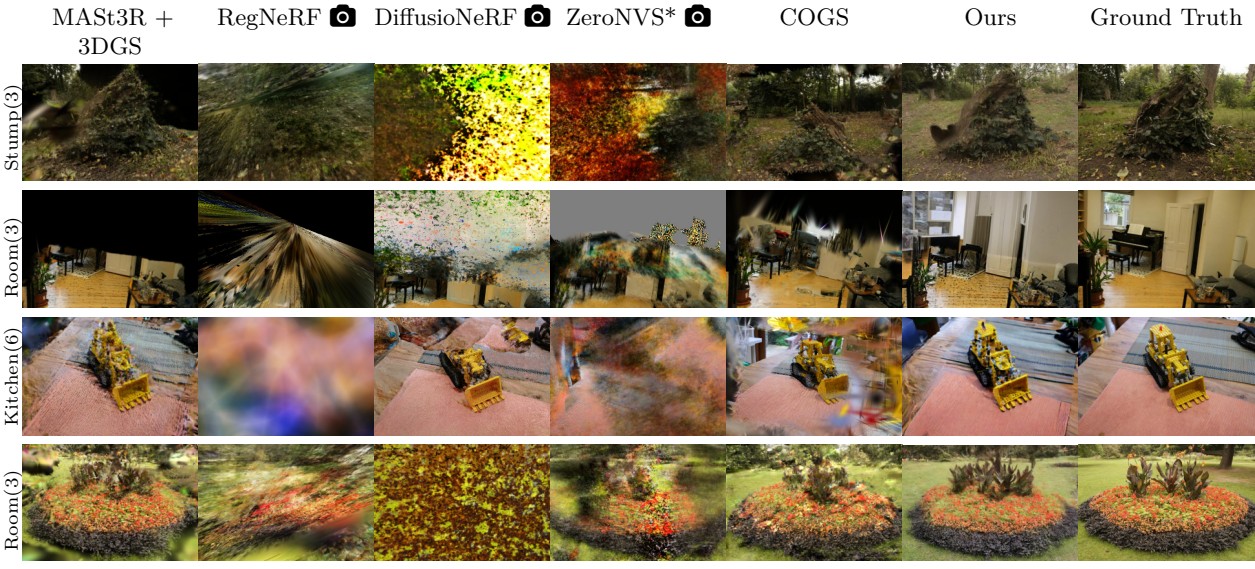

Figure 14: **Additional Qualitative comparisons** on the MipNeRF360 dataset.

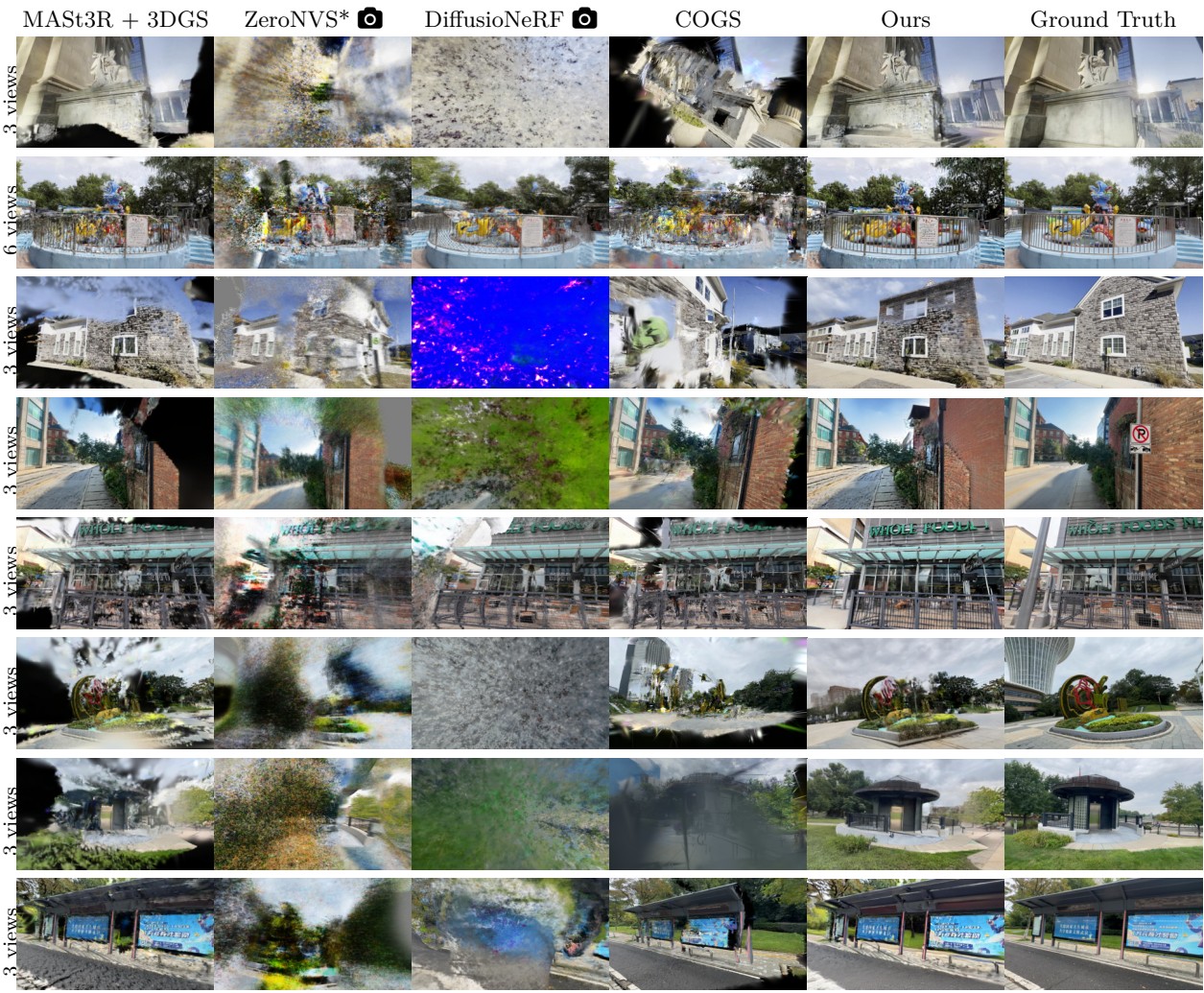

Figure 15: **Additional Qualitative comparisons** on the DL3DV benchmark.

Table 3: **Ablation study**. We report average performance across 3-view splits of *bonsai* and *treehill* scenes on 6 metrics. Our proposed diffusion model variant with conditioning from clip features, plücker embeddings, and confidence maps achieves identical FID, KID, and DISTS scores with a more parameter-heavy cross-attention variant. Red, orange, and yellow indicate best, second-best, and third-best performing methods, respectively.

| Method | PSNR ↑ | LPIPS ↓ | SSIM ↑ | FID ↓ | KID ↓ | DISTS ↓ |
|---|---|---|---|---|---|---|
| Base | 12.024 | 0.271 | 0.574 | 264.562 | 0.105 | 0.272 |
| Base + Conf. | 11.529 | 0.261 | 0.572 | 249.204 | 0.099 | 0.274 |
| w/o CLIP | 12.166 | 0.312 | 0.578 | 177.707 | 0.059 | 0.279 |
| w/o Conf. | 12.100 | 0.270 | 0.572 | 265.564 | 0.131 | 0.271 |
| Cross-Attn | 12.740 | 0.298 | 0.543 | 152.961 | 0.042 | 0.225 |
| MASt3R + 3DGS | 11.641 | 0.269 | 0.604 | 335.064 | 0.268 | 0.329 |
| *GScenes* | 12.871 | 0.297 | 0.545 | 154.769 | 0.039 | 0.228 |

poses instead of ground truth COLMAP poses (posed setting) and show the difference in NVS quality on test views in Fig 16. From this experiment, one can conclude that NVS quality achieved with posed techniques does not directly translate to similar performance in a pose-free setting.

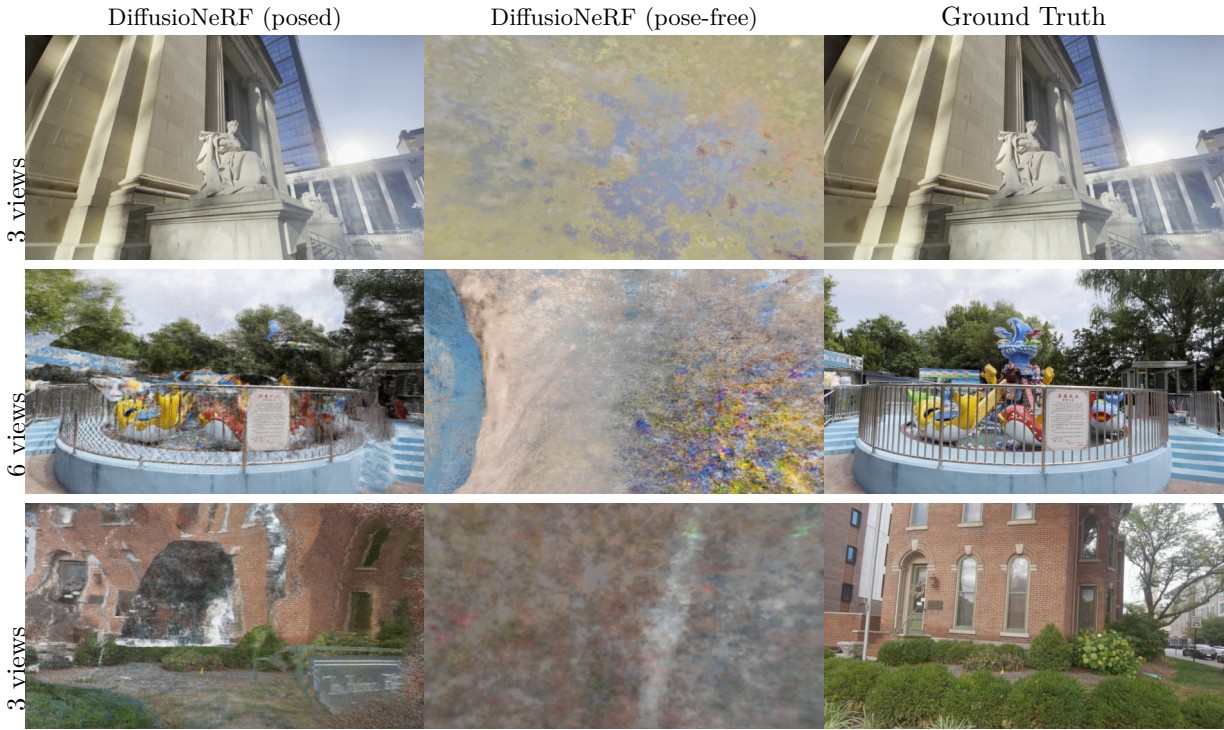

Figure 16: **DiffusioNeRF in a pose-free setting.** We evaluate DiffusioNeRF using poses estimated by MASt3R instead of ground truth colmap poses estimated from dense views. The drop in NVS quality demonstrates that a posed reconstruction technique cannot be trivially adapted to the pose-free setting.

## H  Effect of Generative Priors on Training Views

Table 4: **Effect of Generative Priors on Training Views.** We pick the *Square* scene from DL3DV-10K and its 3, 6, 9 view splits to compare reconstruction performance of *GScenes* with our baseline MASt3R + 3DGS on the training views. Noise and 3D inconsistencies introduced from the generated novel views cause a minor drop-off in the reconstruction quality of training views.

|  | PSNR ↑ | | | SSIM ↑ | | | LPIPS ↓ | | |
|---|---|---|---|---|---|---|---|---|---|
|  | 3-view | 6-view | 9-view | 3-view | 6-view | 9-view | 3-view | 6-view | 9-view |
| MASt3R + 3DGS | 23.34 | 22.32 | 22.96 | 0.73 | 0.62 | 0.63 | 0.20 | 0.21 | 0.23 |
| *GScenes* | 22.44 | 21.52 | 21.70 | 0.75 | 0.57 | 0.56 | 0.18 | 0.25 | 0.26 |

In this section, we analyze the effect of using generative priors for novel view synthesis on the reconstruction quality of the sparse training views of a 3D scene. To ensure 3D consistency and limited artifacts in the reconstructed scene, our diffusion model integrates camera, context, and confidence map conditioning, and the autoregressive 2D-3D distillation process (Sec 2.6) progressively resolves inconsistencies across different novel views. The differences that cannot be reconciled result in artifacts like floaters, which also negatively impact the reconstruction quality of the input views. In Tab 4 and Fig 17, we pick the *Square* scene from DL3DV-10K benchmark and its 3, 6, and 9 view splits to compare the performance of *GScenes* with our

| MASt3R + 3DGS (3) | *GScenes* (3) | MASt3R + 3DGS (6) | *GScenes* (6) | MASt3R + 3DGS (9) | *GScenes* (9) | Ground Truth |
|---|---|---|---|---|---|---|

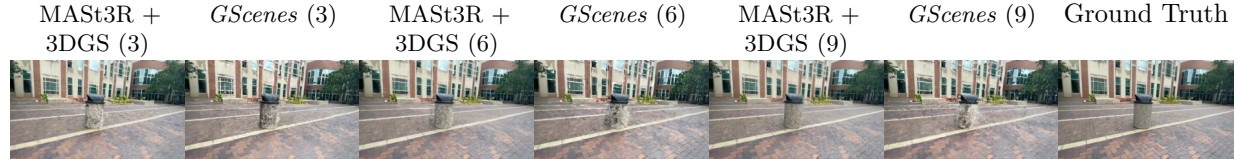

Figure 17: **Effect of Generative Priors on Training Views.**

baseline reconstruction engine - MASt3R + 3DGS on reconstruction of training views. As we can observe, while there is minor degradation both qualitatively and quantitatively due to distillation of generative priors, the original 3DGS objective for training views during autoregressive scene reconstruction ensures that the reconstructed sparse input views still faithfully represent the 3D scene.

# I  Limitations & Future Work

Table 5: **Pose estimation accuracy with *GScenes***. SfM-poses for training views estimated from the full observation set are used as ground truth. We report errors in camera rotation and translation using Absolute Trajectory Error (ATE) and Relative Pose Error (RPE) as in  Bian et al. (2023). Despite the MASt3r initialization of camera extrinsics and subsequent pose optimization, there are large errors w.r.t COLMAP poses of the dense observation set, harming test-pose alignment and subsequent NVS quality.

|  | $\text{RPE}_t \downarrow$ | | | $\text{RPE}_r \downarrow$ | | | ATE $\downarrow$ | | |
|---|---|---|---|---|---|---|---|---|---|
|  | 3-view | 6-view | 9-view | 3-view | 6-view | 9-view | 3-view | 6-view | 9-view |
| *GScenes* | 37.684 | 27.893 | 30.916 | 124.501 | 58.915 | 40.424 | 0.261 | 0.294 | 0.266 |

*GScenes* is a first step towards a generative solution for pose-free sparse-view reconstruction of large complex scenes. However, it is not free of its fair share of limitations. The quality of the final reconstruction depends heavily on the initial relative pose estimation by the MASt3r pipeline, and even though the poses are further optimized jointly with Gaussian attributes during training, there are still large differences with the ground truth COLMAP poses estimated from dense views as we show in Tab 5. This limits fair comparison with posed reconstruction methods, as even the test-time pose alignment step (Sec 3.5) cannot compensate for the initial errors in the pipeline. Our diffusion model, much like related methods, is not agnostic to the 3D representation from which novel view renders, depth, and confidence maps are obtained for fine-tuning the diffusion model for 3D-aware sparse-view NVS. Moreover, our diffusion model trained with 3DGS renders with MASt3R initialization would not be able to rectify novel views from a 3DGS representation with SfM initialization due to the slight difference in the distribution of rendered images. To ensure multiview consistency across all synthesized views, we employ view conditioning in the form of plucker embeddings and use a fixed noise latent across all novel views for multistep reconstruction. However, this does not alleviate the multiview consistency issue completely as novel views are synthesized in an autoregressive manner and not simultaneously synthesized like in video diffusion models. We aim to address these limitations of our diffusion model in future work.

