# OpenReview forum: "Gaussian Scenes: Pose-Free Sparse-View Scene Reconstruction using Depth-Enhanced Diffusion Priors"
_TMLR — Accepted by TMLR_

### Review · Reviewer_ajat · 2025-05-01

**Summary Of Contributions:**

This paper presents a method for 3D reconstruction from sparse views, without assuming access to camera parameters (i.e. it is pose-free). It fits gaussian splats (after MASt3R initialisation) to the input images, then renders from a novel viewpoint, and uses a conditional diffusion model to predict a 'tidied' version of this (RGB+D) rendering, which is then incorporated as pseudo-ground-truth in a further stage of splat optimisation. This process proceeds iteratively, adding more diffusion-generated views. The conditional diffusion model itself is fine-tuned from Stable Diffusion to support the relevant conditioning and RGB+D output, using a relatively small dataset. The method is evaluated on standard datasets of natural scenes, where it outperforms other pose-free methods.

**Audience:**

Yes

**Broader Impact Concerns:**

This work does not raise any specific concerns.

**Claims And Evidence:**

Yes

**Requested Changes:**

There is some redundancy between 2.0 and the 2nd paragraph of 2.1; it might be worth merging these

It would be good if there were a related work section in the main paper rather than just an appendix

The "Generative Priors for NVS" paragraph of related work is missing references to diffusion models that incorporate explicit latent 3D – notably Viewset Diffusion [Szymanowicz, ICCV 2023] and Denoising Diffusion via Image-Based Rendering [Anciukevicius, ICLR 2024]

"competitive" in fig 13 caption should perhaps be weakened – the proposed method's results are consistently worse than CAT3D

**Strengths And Weaknesses:**

**Strengths**

The underlying idea of performing sparse-view reconstruction by iteratively incorporating diffusion-generated views is simple and elegant

The specific approach contains some novel aspects, such as the approach to confidence estimation used in deciding how to weight information, and using RGB+D conditioning and output in the diffusion model (unlike ReconFusion/etc. which use RGB only)

Quantitative results show the method exceeds the performance of two other pose-free approaches – just using MASt3R then gaussian splatting, and the recent COGS – on the standard reconstruction metrics (PSNR, SSIM, LPIPS) across two datasets (MipNeRF-360 and DL3DV)

Qualitative results are significantly better than a 'naive' baseline of MASt3R+3DGS, with plausible completion of partially-observed regions.

There is an ablation study measuring the benefit of several design decisions relating to conditioning in the diffusion model, giving some insight into what information is helpful to the model

There is a balanced discussion of limitations (albeit in an appendix)

Overall the text is clear and readable; the paper is well structured; the figures are well-presented and appropriate

**Weaknesses**

The technical novelty is fairly small – the proposed pipeline is rather reminiscent of ReconFusion, among others. While the pose-free setting differentiates it somewhat, there is very little in the method that is specific to this setting (only the MASt3R initialisation – which itself is not novel), so motivating the work on this basis is rather strange.

Qualitative results are somewhat weak – while the model synthesises 3D-consistent and somewhat plausible content in unobserved regions, this is lacking in detailed texture and notably weaker than e.g. CAT3D.

While it is true that CAT3D and ReconFusion assume known camera poses, these could be estimated using Mast3R or similar; it is unclear whether such a pipeline would in fact perform better or worse (with perfect poses, they are apparently very good; with estimated ones, unknown). In particular, the quantitative and qualitative gap between the proposed method and CAT3D is large enough that it is plausible using it with estimated poses would still outperform the proposed method.

---

> ### Author Response · Authors · 2025-05-20
> **Response to Reviewer ajat**
>
> > Technical Novelty
>
> While our method shares similarities with ReconFusion in utilizing image-conditioned diffusion models for novel-view inference, our approach differs fundamentally in several key aspects. Specifically, our method performs autoregressive scene generation where we rely on the underlying 3D representation to render novel views with empty regions and Gaussian artifacts, and the diffusion model generates a clean image conditioned on this rendered view. ReconFusion, on the other hand, generates pseudo ground truths for novel views conditioned on CLIP features of source views, pixelNeRF features of both source and target views, and uses the NeRF-rendered image only for loss optimization. Importantly, ReconFusion’s diffusion model is dependent on precise ground-truth poses during both training and inference. Since such accurate poses are unavailable in our pose-free scenario, we condition our diffusion model on noisy poses estimated from sparse views by MASt3R. Additionally, we introduce two core technical contributions unique to our work: (1) a confidence map for effectively identifying empty regions and Gaussian artifacts in novel views, and (2) lightweight FiLM modulation layers that significantly enhance context and view conditioning. Our extensive ablation studies further highlight the individual importance of these conditioning elements in achieving high-quality novel view synthesis.
>
> > Weak Qualitative results
>
> We would like to reiterate that both ReconFusion and CAT3D are closed-source posed methods for sparse-view reconstruction. Neither their code nor detailed training procedures have been made publicly available, making independent verification of their reported results impossible. They are only included in our paper for qualitative comparison and to have an upper bound in terms of rendering quality that could be achieved with stronger video diffusion priors and more accurate pose-estimation methods (than MASt3R, DUSt3R) from sparse inputs. We could have performed a fair comparison if we had access to the data, code, and the generative models used in Reconfusion and CAT3D. Even after reaching out to the authors and making our best efforts to validate and reproduce their methodology, the lack of sufficient details in the paper and the use of closed-source data meant we were not able to verify their experiments. We hope that code and models for such methods, when released in the future, will allow a fairer comparison with our paper using open-source code and models. As we show in our experiments, we outperform COGS and MASt3R+3DGS - the 2 most relevant pose-free baselines for our task, and also all open-source posed techniques. ZeroNVS, for example, is one such posed reconstruction method among the baselines, which uses stronger generative priors to aid reconstruction, but GScenes performs better both qualitatively and quantitatively. We have further improved our qualitative results using the densification strategy from Gaussian-MCMC [2] and would encourage all reviewers to check the updated supplementary. While there is certainly room for improvement, as we discuss in our limitations, GScenes is a first step towards solving this challenging problem.
>
> > ReconFusion / CAT3D in pose-free setting
>
> Since it is not possible to evaluate ReconFusion / CAT3D in pose-free setting, we instead evaluate an open-source posed method in DiffusioNeRF from our baselines. This experiment is introduced in Sec. G in the appendix of the updated paper. We show that NVS results with DiffusioNeRF deteriorate substantially when the training poses are estimated by MASt3R from sparse inputs instead of COLMAP from dense inputs. This shows that posed reconstruction techniques cannot be trivially adapted for our pose-free reconstruction task.
>
> > Redundancy between Sec 2.0 and 2.1 in method section
>
> We have updated this in the current version to remove redundancies as much as possible.
>
> > Related Work in main paper
>
> We wanted to include Related Works in the main paper, but a detailed discussion was not possible given page limit constraints.
>
> > Missing references
>
> We have cited Viewset Diffusion and Denoising Diffusion via Image-Based Rendering in the latest version.
>
> > Fig 13 Caption
>
> We have made the caption more accurate in agreement with the visual results.

---

### Review · Reviewer_LtUC · 2025-05-01

**Summary Of Contributions:**

The submission presents a method, GScenes, for pose-free 3D reconstruction of 360° scenes from sparse, uncalibrated images. The approach combines a 3D Gaussian Splatting framework with an RGBD image-to-image diffusion model to refine synthesized novel views. Key technical components include a FiLM-based conditioning mechanism for context and geometry, a pixel-aligned confidence measure to identify unreliable regions, and an iterative reconstruction strategy integrating synthesized views. Experimental results on standard benchmarks show that the method achieves competitive performance with state-of-the-art pose-free and some posed methods, using fewer views and lower computational resources.

**Audience:**

Yes

**Claims And Evidence:**

Yes

**Requested Changes:**

It is necessary to provide a more practical justification for the motivation of the method and to complete the experiments mentioned in the weaknesses.

**Strengths And Weaknesses:**

This paper attempts to address the problem of scene reconstruction under sparse view conditions. Leveraging generative priors to aid the reconstruction is a great idea. The writing and structure of the paper are also clear and easy to follow.


However, there are some shortcomings:
although the authors claim to have achieved state-of-the-art performance, the reconstruction quality is still not very good based on both the quantitative metrics and the visual results. In particular, the rendered images contain a large number of artifacts and do not align well with the real scenes. This raises the question of the method’s motivation. If the goal is to improve rendering for visualization purposes, the current results are not yet at a practically usable level. If the goal is 3D reconstruction, the integration of diffusion models has made the results unreliable.


During training, the newly added novel views might be close to the test views. Could this lead to inaccurate results during testing?

The authors did not show whether the final optimized training camera poses deviate from the ground truth. In the proposed algorithm, newly generated images are added to the training set, which could potentially affect the result of joint camera parameter optimization.

---

> ### Author Response · Authors · 2025-05-20
> **Response to Reviewer LtUC**
>
> > Artifacts in Visual Results:
>
> We would like to emphasize that reconstructing 360 scenes (which typically require several hundred images for high fidelity reconstruction) from extremely sparse unposed inputs like 3, 6, and 9 input images is a challenging and ill-posed problem with potential applications in AR/VR and game development. Generative priors can help infer plausible details in unobserved regions, but their 3D outpainting abilities are also limited, which accounts for poor performance on pixel-aligned metrics like PSNR. Nevertheless, we outperform all open-source posed and pose-free baselines in the reconstruction of complex 360 scenes. Regarding the submitted visual results, we were previously relying on the default densification strategy in 3DGS, which clones or splits existing scene Gaussians based on viewspace positional gradients - something that has been shown to be suboptimal in follow-up papers like Taming 3DGS[1] and Gaussian-MCMC[2]. We have adopted the densification strategy in Gaussian MCMC for our latest video submission, where the presence of artifacts has been substantially curtailed. We would encourage the reviewers to check them in the updated supplementary material (under data/).
>
> > the integration of diffusion models has made the results unreliable.
>
> We are not entirely sure what the reviewer meant by this comment. Supposing that the comment points towards synthesized novel views not aligning with the ground truth images, we would like to emphasize that our generative priors can only ensure that the reconstructed scene remains faithful to the sparse training images, which is why metrics like FID, KID are more appropriate for this problem setting as they measure alignment of distributions of predicted and ground truth test views.
>
> > During training, the newly added novel views might be close to the test views. Could this lead to inaccurate results during testing?
>
> We are not entirely sure about this comment; it would be good if the reviewer could clarify a bit more. We would encourage the reviewer to refer to Section 2.7 for details on test-time pose alignment. We perform a least-squares rigid point set registration to estimate a transformation that maps the test coordinate frame to that of the training reconstruction. On applying this transformation matrix to the test camera poses, test camera extrinsics are expressed in the same coordinate system as the training poses and point cloud for subsequent evaluation. However, this also does not lead to perfect alignment, which is why we use a photometric loss between the rendered and ground truth test images to further optimize the test camera poses, while all 3D Gaussian attributes are kept frozen.
>
> > Optimized poses vs Ground Truth
>
> Since the novel views are estimated along an elliptical trajectory from the optimized train cameras, the training poses are not further optimized when distilling diffusion priors. The cameras estimated by MASt3R are only optimized for the first 1K iterations and then kept fixed for the second stage, as their ground truth images do not change like for novel views. These optimized poses are still erroneous with respect to ground truth COLMAP poses estimated from dense views, as can be seen in Table 4 in the appendix.
>
> ```
> [1] Taming 3DGS: High-Quality Radiance Fields with Limited Resources, Mallick et al., SIGGRAPH Asia 2024
> [2] 3D Gaussian Splatting as Markov Chain Monte Carlo, Kheradmand et al., NeurIPS 2024
> ```

---

> > ### Comment · Reviewer_LtUC · 2025-05-24
> >
> > I would like to clarify the issue I raised regarding the statement "the integration of diffusion models has made the results unreliable." Diffusion models can generate images that may look plausible at first glance, but the shapes or textures of objects in these images often differ from those in the real scene. When such generated images are mixed with sparse real images in the training set, they can introduce multi-view inconsistencies, leading to reconstruction errors such as floaters. This deviates from the goal of accurate real-world reconstruction. Currently, the authors have not evaluated the multi-view consistency of the images generated by the diffusion model, which makes it difficult to support their claim that "the reconstructed scene remains faithful to the sparse training images".

---

> > > ### Comment · Reviewer_LtUC · 2025-05-24
> > >
> > > Regarding the question: “During training, the newly added ...... Could this lead to inaccurate results during testing?”:
> > >
> > > The diffusion model is optimized based on the reconstruction results from sparse views. If there are errors or floaters in the sparse view reconstruction, these issues can also introduce a certain amount of noise into the generated images, as observed in the visual results. If the generated images and the original sparse views are included in the same training set, and the camera position of a generated image is very close to that of one of the sparse views, then the noise in the generated image might affect the sparse view. As a result, the rendering quality of that sparse view could become worse than when only sparse views are used.
> > >
> > > My question is: Have the authors analyzed this issue?

---

> ### Author Response · Authors · 2025-05-29
>
> Thank you for the clarification. Usually, the reconstruction videos are the best way to judge reconstruction quality and multiview consistency of the 3D scene. However, to the best of our knowledge, Met3R [3] is the only recent work in related literature that proposes a metric to evaluate the 3D consistency of a scene. Given $N$ views of a reconstructed scene, this involves computing cosine similarities between DINO features of a pair of views (sampled from $N$) projected onto dense 3D point maps estimated using DUSt3R. This similarity score across all ${N\choose 2}$ pairs is averaged to obtain the final consistency metric - lower the score, better the 3D consistency.
>
> For this experiment, we pick the treehill scene of MipNeRF360 and its 3, 6, and 9 view splits. We choose 10 views sampled from the elliptical trajectory fitted to the training views to evaluate Met3R across 45 pairs of images. The averaged scores for GScenes and all open-source baselines in the paper are reported in the table below. ReconFusion and CAT3D cannot be included as they are closed-source with no available code or implementation.
>
> | Method      | 3-view | 6-view | 9-view |
> |--------------|-------------|-------------|-------------|
> | DiffusioNeRF | 0.538 | 0.528 | 0.574 |
> | ZeroNVS* | 0.626 | 0.586 | 0.593 |
> | COGS | 0.578 | 0.578 | 0.635 |
> | MASt3R + 3DGS | 0.537 | 0.552 | 0.555 |
> | GScenes | **0.496** | **0.513** | **0.479** |
>
> As we can observe, our method achieves the best 3D-consistency score across all baselines and view splits.
>
> In this context, we would like to point out that for sparse-view reconstruction, 3D inconsistencies across different synthesized novel views cannot be eliminated even with view-conditioning heuristics of the diffusion model (e.g., plucker coordinate embeddings in GScenes, pixelNeRF features in ReconFusion, etc.) - we can try to increase consistency by constraining the conditional probability distribution of the generative process through camera and context conditioning. The 2D-3D autoregressive distillation process helps reduce such inconsistencies substantially, but the differences between neighboring views that cannot be reconciled in the final 3D representation result in artifacts like floaters. Video diffusion models are more adept than image-conditioned diffusion models for 3D-consistent NVS (as we discuss in the limitations) since they synthesize multiple novel views simultaneously, but they also cannot eliminate this issue completely.
> ```
> [3] MEt3R: Measuring Multi-View Consistency in Generated Images, Asim et al., CVPR 2025
> ```

---

> > ### Author Response · Authors · 2025-05-30
> >
> > > Effect of Generative Priors on Training Views
> >
> > Thank you for the clarification and the interesting question. We have added a section in the appendix (Sec H) to analyze this issue with a small experiment. Please find it in the updated submission.

---

### Review · Reviewer_3BVF · 2025-05-05

**Summary Of Contributions:**

- The paper proposes a novel, pose-free 360° scene reconstruction pipeline that fuses an initial point cloud generated by MASt3R with a 3DGS optimization stage, followed by diffusion-based inpainting.

- It introduces a lightweight conditioning mechanism using FiLM layers—combining CLIP-based contextual features and Plücker-embedded geometric cues—to achieve performance comparable to cross-attention with fewer than one-third of the parameters.

- A per-pixel confidence map is defined by combining transmittance and Gaussian density, enabling precise detection and correction of reconstruction artifacts.

- On the MipNeRF360 and DL3DV-10K benchmarks, the method achieves state-of-the-art performance among pose-free approaches and approaches the accuracy of pose-aware techniques.

**Audience:**

Yes

**Broader Impact Concerns:**

No concern.

**Claims And Evidence:**

Yes

**Requested Changes:**

Please address the weaknesses outlined above.

**Strengths And Weaknesses:**

**Strengths**
- Novelty: Integrating depth-conditioned diffusion into a pose-free 3D reconstruction framework is an interesting direction. The FiLM-based conditioning elegantly reduces parameter count while preserving reconstruction quality.

- Practicality: The end-to-end pipeline runs in minutes and does not require massive multi-view datasets or high-end GPUs. An open-source implementation will further accelerate adoption.

- Extensive Evaluation: Quantitative results on two major benchmarks (MipNeRF360, DL3DV-10K), along with FID, KID, and DISTS scores, provide a comprehensive assessment of performance.

**Weaknesses**
- Limited Dataset Coverage: Several datasets used by key pose-free baselines—such as RealEstate10K, LLFF, DTU, and CO3D in CAT3D—are missing from the comparison, making it difficult to gauge relative gains.

- Low PSNR Scores: All reported PSNR values fall below 20 dB, raising concerns about the method’s practical usefulness. The teaser images report values below 14 dB, which suggests poor fidelity.

- Lack of Quantitative Ablation for the Confidence Map: While qualitative examples illustrate the confidence map’s utility, no quantitative ablation isolates its contribution to overall performance.

---

> ### Author Response · Authors · 2025-05-20
> **Response to Reviewer 3BVF**
>
> > Limited Dataset Coverage: Several datasets used by key pose-free baselines—such as RealEstate10K, LLFF, DTU, and CO3D in CAT3D—are missing from the comparison, making it difficult to gauge relative gains.
>
> We would like to point out that we perform experimental evaluation at the same scale as all open-source posed and pose-free baselines in our paper. For example, DiffusioNeRF evaluates on 8 scenes of LLFF and 15 scenes of DTU, ZeroNVS evaluates on DTU and MipNeRF360. The most relevant baseline for the pose-free setting in our paper, COGS, reports qualitative and quantitative results on 13 scenes - 8 from Tanks and Temples, and 5 from Static Hikes. Moreover, due to our objective of complex scene reconstruction from few views, our experiments are focused on challenging 360 scenes from DL3DV-10K benchmark and MipNeRF360 datasets, both of which are higher in difficulty than all 3 datasets mentioned here, as noted by previous literature (Tab 1, ReconFusion). Moreover, LLFF and DTU are datasets with forward-facing scenes (no 360 coverage), which lowers the need for generative priors for high-fidelity reconstruction. Our simple MASt3R+3DGS baseline achieves high-quality reconstruction for LLFF scenes. Quantitative evaluation on 2 example scenes is shown below. We also share sample reconstruction videos with MASt3R+3DGS in the updated supplementary material (under data/llff/).
>
> | Scene      | PSNR 3-view | PSNR 6-view | PSNR 9-view | SSIM 3-view | SSIM 6-view | SSIM 9-view | LPIPS 3-view | LPIPS 6-view | LPIPS 9-view |
> |--------------|-------------|-------------|-------------|-------------|-------------|-------------|---------------|---------------|---------------|
> | Fern | 18.22 | 18.68 | 21.79 | 0.56 | 0.72 | 0.74 | 0.22 | 0.18 | 0.12 |
> | Trex | 16.80 | 20.44 | 20.06 | 0.60 | 0.70 | 0.75 | 0.22 | 0.14 | 0.15 |
>
> | Scene      | FID 3-view | FID 6-view | FID 9-view | KID 3-view | KID 6-view | KID 9-view | DISTS 3-view | DISTS 6-view | DISTS 9-view |
> |--------------|-------------|-------------|-------------|-------------|-------------|-------------|---------------|---------------|---------------|
> | Fern | 80.36 | 42.88 | 34.55 | 0.02 | 0.01 | 0.01 | 0.12 | 0.08 | 0.06 |
> | Trex | 120.34 | 124.80 | 77.75 | 0.08 | 0.09 | 0.02 | 0.13 | 0.11 | 0.09 |
>
> > Low PSNR Scores: All reported PSNR values fall below 20 dB, raising concerns about the method’s practical usefulness. The teaser images report values below 14 dB, which suggests poor fidelity.
>
> As noted in prior work (e.g., Chan et al., 2023; Sargent et al., 2024) and discussed in Section 3.1 of our paper, pixel-level metrics such as PSNR and SSIM are not reliable indicators of performance for generative novel view synthesis methods. These models aim to produce plausible reconstructions that are perceptually consistent with the scene, but not necessarily identical to the ground truth at the pixel level. Consequently, PSNR values may appear low even when the reconstructed views are visually faithful. Similar PSNR values are also reported in recent literature, including ZeroNVS (Sargent et al., 2024), which performs novel view synthesis from a single input image. We emphasize that PSNR and SSIM are reported primarily for completeness. For a more meaningful assessment, we urge the reviewers to consider perceptual and distributional metrics such as LPIPS, FID, KID, and DISTS. On these, our method performs competitively with the closed-source ReconFusion on LPIPS, and surpasses all open-source baselines, on FID, KID, and DISTS.
>
> ___________________
> > Lack of Quantitative Ablation for the Confidence Map: While qualitative examples illustrate the confidence map’s utility, no quantitative ablation isolates its contribution to overall performance.
>
> Please find the quantitative performance corresponding to Fig. 11 in Table 3 in the Appendix. Without the confidence map conditioning (Row 4), performance is similar to the Base variant (Row 1) with only Img-Img diffusion, even with context and geometry conditioning. Our proposed confidence map complements context and geometry conditioning using both cross-attention (Row 5) and FiLM modulation layers (Row 7).

---

### Author Response · Authors · 2025-05-20
**Rebuttal Summary**

We thank the reviewers for highlighting our method's novelty (Reviewer 3BVF), simplicity and elegance (Reviewer ajat), and interesting direction in integrating generative priors into sparse-view, pose-free 3D reconstruction (Reviewer 3BVF), as well as recognizing our method’s extensive evaluation (Reviewer 3BVF), and clear and readable presentation (Reviewers LtUC, ajat). Regarding dataset coverage, we clarify that our evaluation meets or exceeds standards set by open-source baselines (DiffusioNeRF, ZeroNVS, COGS), and we additionally provide experiments on 2 LLFF scenes with our MASt3R+3DGS baseline to validate our claims. Addressing the concerns about low PSNR scores, we point reviewers toward established literature showing limited suitability of pixel-level metrics for generative methods, emphasizing instead our strong performance on perceptual and distributional metrics (LPIPS, FID, KID, DISTS). To tackle visual artifacts, we have substantially improved the quality of our reconstruction results by adopting the densification strategy from Gaussian-MCMC, demonstrated in the new supplementary videos. Finally, we reiterate the reproducibility and transparency advantage of our open-source approach over closed-source methods (ReconFusion, CAT3D).

---

### Decision · Action_Editor_fRtV · 2025-06-11

**Recommendation:** Accept as is

**Additional Comments:**

typo:  Reconfusion-->ReconFusion

**Audience:**

Yes

**Audience Explanation:**

The paper offers a new approach for scene reconstruction from few unposed images.  As the reviewers mention, the paper offers an elegant approach for few-view reconstruction from unposed images by incrementally synthesizing and incorporate new views. The general topic and specific tools and innovations will be of interest to many.

**Claims And Evidence:**

Yes

**Claims Explanation:**

The method is evaluated and ablated on the MipNERF360 and DL3DV-10K datasets. PSRN results appear unfavorable to the method, but it is good that they are included, and the authors explain the limitations of the PSNR metric. While more extensive evaluation would be possible, the current experiments meet the standards set by other related open source works and are sufficient to validate the claims.  Limitations are sufficiently discussed.

Two of the three reviewers recommend that the paper is accepted. They express limited enthusiasm for the novelty of the method and quality of results, but that the paper meets the TMLR criteria for acceptance. A third reviewer acknowledges the paper's technical contributions but recommends against acceptance, on the basis that it's not easy to identify a clear use case or niche where the method has a clear advantage. I agree with the majority recommendation that the claims are sufficiently supported by evidence.